# Bayesian and Non-Bayesian Inference for Unit-Exponentiated Half-Logistic Distribution with Data Analysis

Amal S. Hassan [1,*], Aisha Fayomi [2], Ali Algarni [2] and Ehab M. Almetwally [3,4,*]

1    Faculty of Graduate Studies for Statistical Research, Cairo University, Giza 12613, Egypt
2    Faculty of Science, Department of Statistics, King Abdulaziz University, Jeddah 21589, Saudi Arabia
3    Faculty of Business Administration, Delta University of Science and Technology, Gamasa 11152, Egypt
4    The Scientific Association for Studies and Applied Research, Al Manzalah 35646, Egypt
*    Correspondence: amal52_soliman@cu.edu.eg (A.S.H.); ehab.metwaly@deltauniv.edu.eg (E.M.A.)

**Abstract:** Unit distributions are typically used in probability theory and statistics to illustrate useful quantities with values between zero and one. In this paper, we investigated an appropriate transformation to propose the unit-exponentiated half-logistic distribution (UEHLD), which is also beneficial for modelling data on the unit interval. This distribution's mathematical features are supplied, including moments, probability-weighted moments, incomplete moments, various entropy measures, and stress–strength reliability. Using well-known estimation techniques such as the maximum likelihood, maximum product of spacing, and Bayesian inference, the estimators of the parameters relevant to the proposed distribution were determined. A comprehensive simulation analysis is provided to examine the performance of parameter estimation approaches on finite samples. The proposed distribution was realistically applied to data on economic growth and data on the tensile strength of polyester fibers to provide an explanation. Furthermore, the analysis of COVID-19 data from Britain as a medical statistical dataset is provided. The experimental results demonstrate that the suggested UEHLD yields a better comparison with some new unit distributions, as well as other unbounded distributions.

**Keywords:** unit distributions; exponentiated half-logistic distribution; moments; entropy measures; quantile; Bayesian method; COVID-19 data

**MSC:** 62E15; 62F15; 62F03; 46N10



## 1. Motivation and Introduction

In recent times, the construction of bounded distributions has grown tremendously. The unit distributions are necessary for modelling proportions, scores, rates, indices, etc. The benefit of these bounded distributions is that they allow the basic distribution to be more flexible throughout the unit interval without introducing new parameters. The well-known beta distribution is perhaps the first model that was employed for data observed in the interval (0, 1). Besides this, Kumaraswamy's distribution (Kumaraswamy [1]) and the Topp–Leone distribution (Topp and Leone [2]) deserve attention.

Recently, several notable models have been presented as alternatives to the beta and Kumaraswamy distributions in order to improve modelling flexibility using appropriately transformed methods. The transformation of the well-known continuous distributions has frequently been used to present the newly suggested unit distributions. The transformations of the form $T = 1/(1 + X)$, $T = X/(1 + X)$, and $T = e^{-X}$, are popularly used when $X$ is a positive-valued random variable. Several probability distributions have been developed in this area for dealing with restricted datasets in various domains using the previous transformation. Among the most notable distributions are: unit gamma (Grassia [3] and Tadikamalla [4]), unit logistic (Tadikamalla and Johnson [5] and Menezes et al. [6]), unit Birnbaum–Saunders (Mazucheli et al. [7]), unit

Weibull (Mazucheli et al. [8,9]), unit inverse Gaussian (Ghitany et al. [10]), unit Gompertz (Mazucheli et al. [11]), unit Lindley (Mazucheli et al. [12]), unit generalized half-normal (Korkmaz [13]), unit modified Burr-III (Haq et al. [14]), unit Burr-XII (UBXII) (Korkmaz and Chesneau [15]), unit gamma/Gompertz (Bantan et al. [16]), unit log-logistic (Ribeiro-Reis [17]), and unit generalized log Burr XII (UGLBXII) (Bhatti et al. [18]). In addition to the preceding transformations, Cancho et al. [19] and Rodrigues et al. [20] developed a broad strategy based on the cumulative distribution function (CDF)-quantile function (QF).

In a wide range of academic and professional fields, lifetime modeling and analysis are essential components of statistical work. A complete explanation of an event that only happens once in a lifetime typically follows a thorough data analysis based on a carefully chosen statistical model. The development of new probability distributions was necessary as a result of several attempts to construct models with different features. Lifetime distributions, a crucial statistical technique, can be used to model the numerous properties of lifetime datasets. These datasets can be analyzed using rather complicated distributions found in the statistical literature. One of the most-significant lifetime models is the exponentiated half-logistic distribution (EHLD). It has a number of characteristics that make it a viable alternative to well-known distributions, and at the same time, it has the ability to model various real datasets. It has been used in a variety of fields, including insurance, engineering, medical, and education. The probability density function (PDF) and CDF of the EHLD are represented by:

$$f(y) = \frac{2\delta\varphi e^{-\delta y}}{\left(1 + e^{-\delta y}\right)^2} \left(\frac{1 - e^{-\delta y}}{1 + e^{-\delta y}}\right)^{\varphi - 1} ; \quad y, \delta, \varphi > 0, \tag{1}$$

$$F(y) = \left(\frac{1 - e^{-\delta y}}{1 + e^{-\delta y}}\right)^{\varphi} ; \quad y, \ \delta, \varphi > 0,$$

where $\delta$, and $\varphi$ are the scale and shape parameters, respectively. PDF (1) gives the half-logistic distribution (HLD) for $\varphi = 1$. The EHLD has recently caught the interest of a large number of academics. The parameters and reliability estimators of the EHLD utilizing the maximum likelihood (ML) and Bayesian techniques were examined using a progressive censoring scheme [21–23]. Cordeiro et al. [24] enhanced the EHLD by proposing the notion of the EHLD as a generator to produce the family of continuous distributions with the goal of making the distributions more practical. Seo and Kang [25] examined the moment and ML estimators of the EHLD parameters. EHLD's ML, inverse moment, and modified inverse moment estimators, as well as the joint confidence regions were investigated by Gui [26]. Naidu et al. [27] created a reliability test strategy for the EHLD. Jeon and Kang [28] used multiple Type-I hybrid censoring to look at estimators of the EHLD parameters. Adaptive progressive censoring was used by Xiong and Gui [29] to investigate the parameter estimators of the EHLD.

The goal of this essay is to construct a new probability, called the unit-EHLD (UEHLD) by using transformation $T = e^{-Y}$, where $Y$ is the EHLD. We were motivated to propose the UEHLD due to the following:

(i)   To create various forms for the hazard rate function (HRF) and PDF.
(ii)  With a range of 0 to 1, the UEHLD is versatile and may be used to describe a variety of datasets. It can be seen as a useful model for fitting skewed data that might not be effectively fit by other popular distributions.
(iii) To give a comprehensive comparison of three approaches for estimating the UEHLD parameters, as well as an examination of the performance of such estimators for various parameter values and sample sizes. Our investigations were limited to the ML, maximum product of spacing (MPS), and Bayesian methods. It is difficult to theoretically examine the behaviors of different estimating approaches; thus, we carried out extensive simulation studies to evaluate the behaviors of different estimators with the bias, mean-squared error (MSE), and length of the confidence interval (CI) criteria.

(iv) To describe some practical uses in many disciplines, such as economic growth data, tensile strength data, and COVID-19 data.

This article's structure is as follows: In Section 2, we develop a brand-new model known as the UEHLD. Some moments' measures are deduced in Section 3. The stress–strength model and some information measures are described in Section 4. Section 5 discusses a few of the various techniques for estimating the model parameters. Section 6 carries out a numerical investigation using Monte Carlo simulations. Section 7 conducts a numerical examination of real datasets, and Section 8 gives the findings.

## 2. Unit-Exponentiated Half-Logistic Distribution

In this section, the CDF and PDF of the UEHLD are proposed. The HRF and QF of the UEHLD are provided.

**Definition 1.** *Let Y be a random variable having the EHLD, with parameters δ, and φ and $T = e^{-Y}$, then the CDF of the bounded UEHLD with support on (0, 1) is formed as:*

$$G(t) = P(T \leq t) = P\left(e^{-Y} \leq t\right) = 1 - P(Y \leq -\ln t) = 1 - F_Y(-\ln t),$$

*which eventually leads to*

$$G(t) = 1 - \left(\frac{1 - t^\delta}{1 + t^\delta}\right)^\varphi; \delta, \varphi > 0, \ 0 < t < 1. \tag{2}$$

From (2), we have $G(t) = 0$, for $t \leq 0$, and $G(t) = 1$, for $t \leq 1$. The PDF of the UEHLD is represented by:

$$g(t) = \frac{2\varphi\delta t^{\delta-1}}{(1 + t^\delta)^2}\left(\frac{1 - t^\delta}{1 + t^\delta}\right)^{\varphi-1}; \qquad \delta, \varphi > 0, \qquad 0 < t < 1. \tag{3}$$

For $\varphi = 1$, PDF (3) provides a UHLD. The UEHLD density is shown in Figure 1 in a variety of forms, including right-skewed, left-skewed, revers-J, U-shaped, and asymmetric.

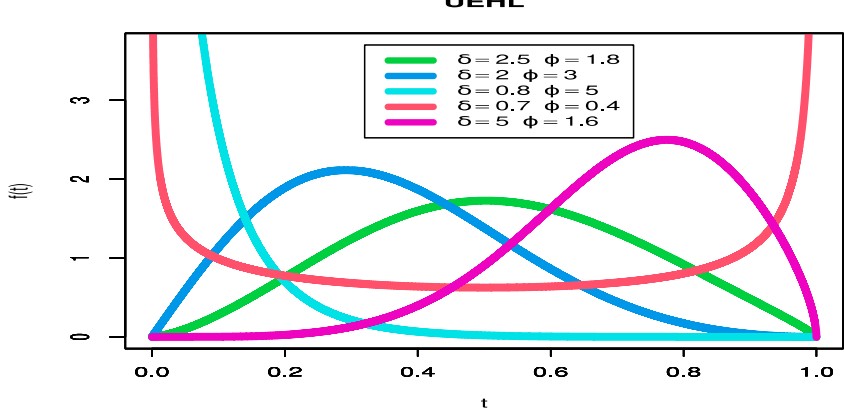

**Figure 1.** Different shapes of the PDF for the unit-exponentiated half-logistic (UEHL) distribution.

The HRF and reverse HRF are as below:

$$h(t) = 2\varphi\delta t^{\delta-1}\left(1 - t^{2\delta}\right)^{-1}$$

$$rh(t) = 2\varphi\delta t^{\delta-1}\left(1 - t^\delta\right)^{\varphi-1}\left(1 + t^\delta\right)^{\varphi+1}\left[1 - \left(\frac{1 - t^\delta}{1 + t^\delta}\right)^\varphi\right]^{-1}.$$

The HRF and its reverse plots of the UEHLD are given in Figure 2 for specific values of the parameters, which can be increasing, decreasing, J-shaped, and bathtub-shaped.

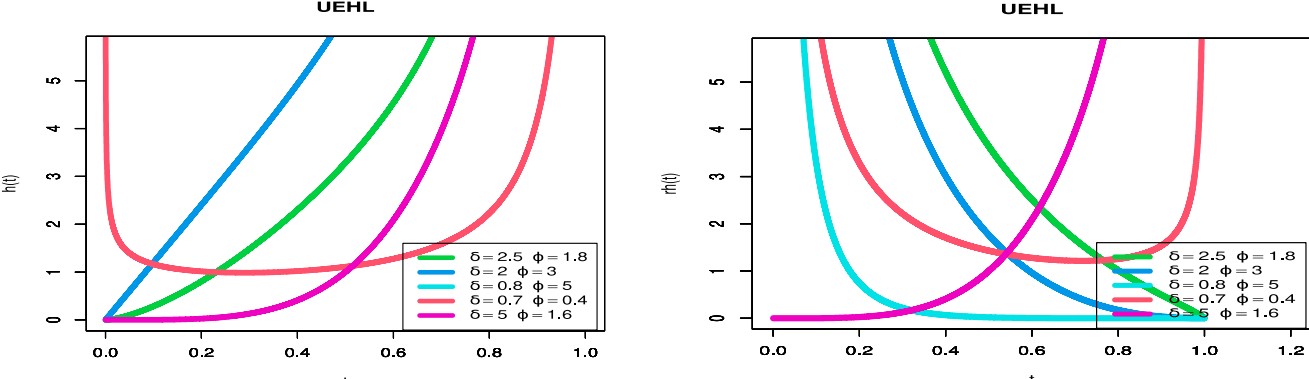

**Figure 2.** Different shapes of the HRF and reverse HRF for the UEHLD.

The QF of the UEHLD is yielded by inverting CDF (2) as follows:

$$Q(u) = G^{-1}(t) = \left( \frac{1 - (1-u)^{1/\phi}}{1 + (1-u)^{1/\phi}} \right)^{1/\delta},$$

(4)

where $u$ is a uniform distribution on (0, 1). We can find the median, upper, and lower quantiles, represented by Q(0.75), Q(0.25), and Q(0.5), by setting $u$ = 0.75, 0.5, and 0.25 in (4).

### 3. Moments and Related Measures

In this section, the moments and other associated measures of the UEHLD such as incomplete moments, mean residual life (MRL), mean inactivity time, and probability-weighted moments (PWMs), are computed.

#### 3.1. Moments and Incomplete Moments

If $T$ has PDF (3), then the $k$th moment of the UEHLD is derived as:

$$\mu'_k = \int_0^1 2\varphi\delta t^{k+\delta-1} \left(1 - t^\delta\right)^{\varphi-1} \left(1 + t^\delta\right)^{-\varphi-1} dt.$$

(5)

Using the binomial expansion in (5), then we have

$$\mu'_k = \sum_{q=0}^\infty (-1)^q 2\varphi\delta \binom{\varphi+q}{q} \int_0^1 t^{k+\delta+\delta q-1} \left(1 - t^\delta\right)^{\varphi-1} dt.$$

(6)

After some simplification, then (6) obtains the form

$$\mu'_k = \sum_{q=0}^\infty (-1)^q 2\varphi \binom{\varphi+q}{q} B\left(\frac{k}{\delta} + q + 1, \varphi\right),$$

where B(.,.) is the beta function. Furthermore, the $k$th central moment of a given random variable $T$ is defined by

$$\mu_k = E(T - \mu'_1)^k = \sum_{i=0}^k (-1)^i \binom{k}{i} (\mu'_1)^i \mu'_{k-i}.$$

Table 1 shows numerical values for the first four moments, variance ($\sigma^2$), skewness ($\alpha_3$), and kurtosis ($\alpha_4$) of the UEHLD.

**Table 1.** Moment measures of the UEHLD.

| $\mu'_k$ | $\varphi = 0.5,$ $\delta = 0.5$ | $\varphi = 0.5,$ $\delta = 1$ | $\varphi = 0.5,$ $\delta = 1.5$ | $\varphi = 1,$ $\delta = 0.5$ | $\varphi = 1.5,$ $\delta = 0.5$ | $\varphi = 1,$ $\delta = 1$ | $\varphi = 1.5,$ $\delta = 1.5$ |
|---|---|---|---|---|---|---|---|
| $\mu'_1$ | 0.429 | 0.227 | 0.137 | 0.571 | 0.655 | 0.386 | 0.401 |
| $\mu'_2$ | 0.31 | 0.121 | 0.055 | 0.429 | 0.51 | 0.227 | 0.217 |
| $\mu'_3$ | 0.255 | 0.082 | 0.031 | 0.356 | 0.429 | 0.159 | 0.137 |
| $\mu'_4$ | 0.221 | 0.062 | 0.02 | 0.31 | 0.376 | 0.121 | 0.096 |
| $\sigma^2$ | 0.126 | 0.07 | 0.036 | 0.103 | 0.081 | 0.078 | 0.056 |
| $\alpha_3$ | 0.292 | 1.241 | 1.934 | −0.205 | −0.494 | 0.486 | 0.41 |
| $\alpha_4$ | 1.569 | 3.471 | 6.429 | 1.664 | 2.017 | 2.093 | 2.287 |

Table 1 demonstrates that, when the value of $\delta$ increases for a fixed value of $\varphi$, we observe that the first four moments and the $\sigma^2$ and $\alpha_3$ measures decrease, while the $\alpha_4$ measure increases. When the value of $\varphi$ increases for a fixed value of $\delta$, we conclude that the first four moments and the $\alpha_4$ measure decrease, while $\sigma^2$ and $\alpha_3$ increase. The distribution is positive-skewed and negative-skewed. Consequently, the UEHLD can be used for modeling both positively and negatively skewed datasets. Furthermore, it is leptokurtic and platykurtic. Figure 3 provides the 3D plots of the mean ($\mu'_1$), $\sigma^2$, $\alpha_3$, $\alpha_4$, coefficient of variation (CV), and index of dispersion (ID).

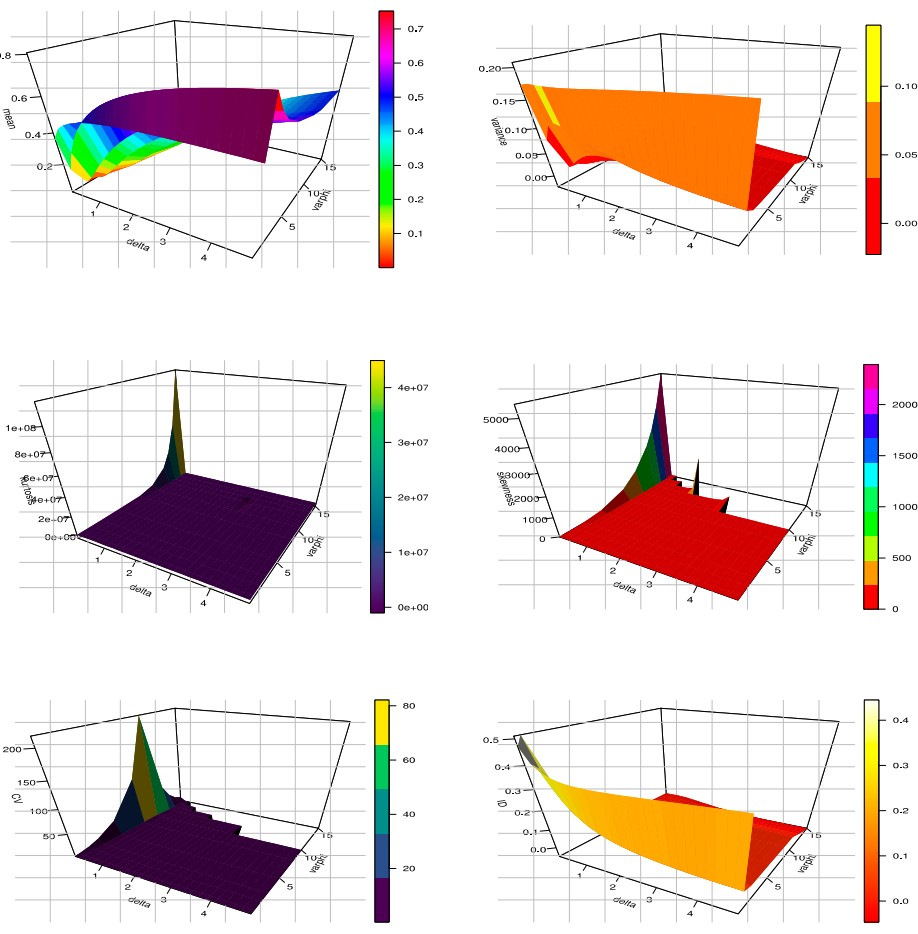

**Figure 3.** The 3D shapes of the mean, $\sigma^2$, $\alpha_3$, $\alpha_4$, CV, and ID for the UEHLD.

The $k$th incomplete moment, say $\xi_k(x)$, of the UEHLD is given by:

$$\xi_k(x) = \int_0^x 2\varphi\delta t^{k+\delta-1}\left(1-t^\delta\right)^{\varphi-1}\left(1+t^\delta\right)^{-\varphi-1}dt.$$

Using binomial expansion and letting $y = t^\delta \Rightarrow dy = \delta t^{\delta-1}dt$, then $\xi_k(x)$ can be written as:

$$\xi_k(x) = \sum_{q=0}^{\infty} (-1)^q 2\varphi\binom{\varphi+q}{q}\mathrm{B}\left(\frac{k}{\delta}+q+1, \varphi, x^\delta\right), \tag{7}$$

where B(.,.,$x$) stands for the incomplete beta function. The first incomplete moment, for $k = 1$, in (7), is used in a variety of ways, including the Bonferroni and Lorenz curves. These curves are commonly utilized in a variety of disciplines.

### 3.2. Residual and Reversed Residual Life Functions

The $m$th moment of the residual life (RL), say $\Lambda_m(x)$, denoted by $\Lambda_m(x) = E\left[(T-x)^m|T>x\right], m = 1, 2, \ldots$, uniquely determines the CDF $G(t)$. The $m$th moment of the RL of $T$ is defined by:

$$\Lambda_m(x) = \frac{1}{\overline{G}(x)}\int_x^1 (T-x)^m\,dG(t). \tag{8}$$

Using the binomial expansion, more than one time in (8), then $\Lambda_m(x)$ of the UEHLD can be expressed as follows:

$$\Lambda_m(x) = \frac{\Xi_{r,q}}{\overline{G}(x)}\int_x^1 \delta t^{r+\delta+\delta q-1}\left(1-t^\delta\right)^{\varphi-1}dt,$$

where, $\Xi_{r,q} = \sum_{r=0}^{m}\sum_{q=0}^{\infty} (-1)^{q+m-r}(x)^{m-r}2\varphi\binom{m}{r}\binom{\varphi+q}{q}$.

Let $y = 1 - t^\delta \Rightarrow dy = -\delta t^{\delta-1}dt$, then $\Lambda_m(x)$ is as below:

$$\Lambda_m(x) = \frac{\Xi_{r,q}}{\overline{G}(x)}\int_0^{1-x^\delta} (1-y)^{\frac{r}{\delta}+q}\,y^{\varphi-1}dy = \frac{\Xi_{r,q}}{\overline{G}(x)}\mathrm{B}\left(\frac{r}{\delta}+q, \varphi, 1-x^\delta\right). \tag{9}$$

The MRL of the UEHLD is the expected value of the remaining lifetimes after duration $t$, which is obtained by putting $m = 1$ in (9). The $m$th moment of the reversed RL, denoted by $\Upsilon_m(x) = E\left[(x-T)^m|T\leq x\right], x > 0, m = 1, 2, \ldots$, is defined by:

$$\Upsilon_m(x) = \frac{1}{G(x)}\int_0^x (x-t)^m dG(t). \tag{10}$$

Using the binomial expansion, more than one time in (10), then $\Upsilon_m(x)$ of the UEHLD can be formed as follows:

$$\Upsilon_m(x) = \frac{1}{G(x)}\sum_{r=0}^{m}\sum_{q=0}^{\infty} (-1)^{m+q}2\varphi\binom{m}{r}\binom{\varphi+q}{q}x^{m-r}\mathrm{B}\left(\frac{m}{\delta}+q+1, \varphi, x^\delta\right).$$

The MIT, also known as the mean reverse RL function, is defined as $\Upsilon_1(x) = E[(x-T)|T\leq x]$, and reflects the amount of time that has passed after an item has failed $(0, x)$. By setting $m = 1$, we obtain the MIT of the UEHLD.

### 3.3. Probability-Weighted Moments

The PWM approach is seen as a generalization of the classical probability distribution moments. The estimators of the parameters and quantiles of generalized distributions may be derived using PWMs. These moments have low variances and no significant biases, and they outperform estimators derived using the ML technique. The PWMs, denoted by $\mathfrak{M}_{a,b}$, for $a \geq 1,\ b \geq 0$, of a random variable $T$, can be designed as:

$$
\begin{aligned}
\mathfrak{M}_{a,b} &= E[T^a G(t)^b] = \int_{-\infty}^{\infty} t^a (G(t))^b g(t) dt. \\
&= \int_0^1 2\varphi \delta t^{a+\delta-1} \left(1 - t^\delta\right)^{\varphi-1} \left(1 + t^\delta\right)^{-\varphi-1} \left[1 - \left(\frac{1-t^\delta}{1+t^\delta}\right)^\varphi\right]^b dt.
\end{aligned}
\tag{11}
$$

The PWMs (11), based on binomial expansion, can be written as:

$$
\begin{aligned}
\mathfrak{M}_{a,b} &= \sum_{u=0}^{b} \sum_{d=0}^{\infty} (-1)^{u+d} \binom{b}{u} \binom{\varphi(u+1)+d}{d} \int_0^1 2\varphi \delta t^{a+\delta(d+1)-1} \left(1-t^\delta\right)^{\varphi(u+1)-1} dt \\
&= \mathrm{K}_{u,d}\, \mathrm{B}\!\left(\tfrac{a}{\delta}+d+1, \varphi+\varphi u\right),
\end{aligned}
$$

where $\mathrm{K}_{u,d} = \sum_{u=0}^{b} \sum_{d=0}^{\infty} (-1)^{u+d} 2\varphi \binom{b}{u} \binom{\varphi(u+1)+d}{d}$, and $\mathrm{B}(.,.)$ is the beta function.

## 4. Stress Strength Reliability and Information Measures

In this section, the stress–strength (S-S) reliability of the UEHLD is provided and some information measures such as the Rényi (Ré), Havrda and Charvat (H-C), and $o-$ entropies are examined.

### 4.1. S-S Parameter

The notion "S-S reliability", denoted by $\Re = P[T_2 < T_1]$, is illustrated by $T_1$ representing component strength and $T_2$ representing stress. If $T_2 > T_1$ in S-S modeling, the system is a mail function. Let $T_1 \sim$ UEHLD $(\delta, \varphi_1)$ and $T_2 \sim$ UEHLD $(\delta, \varphi_2)$, then $\Re$ is determined as follows:

$$
\Re = 1 - \int_0^1 \frac{2\varphi_1(\varphi_1+\varphi_2)\delta t^{\delta-1}}{(\varphi_1+\varphi_2)(1+t^\delta)^2} \left(\frac{1-t^\delta}{1+t^\delta}\right)^{\varphi_1+\varphi_2-1} dt = \frac{\varphi_2}{(\varphi_1+\varphi_2)}.
\tag{12}
$$

The S-S reliability in (12) depends on $\varphi_1$, and $\varphi_2$.

### 4.2. Information Measures

Entropy measures the presence of uncertainty or variability in a random variable. The higher the entropy number, the more uncertain the data are. This section focuses on determining the UEHLD expression for various entropy measurements. The Ré entropy of a random variable $T$ is mathematically specified by:

$$
\mathbb{R}(\zeta) = (1-\zeta)^{-1} \log\left(\int_0^\infty (g(t))^\zeta dt\right), \zeta \neq 1, \zeta > 0.
\tag{13}
$$

Substituting PDF (3) in (13) and using binomial expansion, then the Ré entropy of the UEHLD is

$$
\mathbb{R}(\zeta) = (1-\zeta)^{-1} \log\left(\sum_{q=0}^{\infty} (-1)^q \binom{\zeta(\varphi+1)+q}{q} (2\varphi\delta)^\zeta \int_0^1 t^{\zeta(\delta-1)+\delta q} \left(1-t^\delta\right)^{\zeta(\varphi-1)} dt\right).
$$

Suppose that $y = t^\delta \Rightarrow dy = \delta t^{\delta-1} dt$, then $\mathbb{R}(\zeta)$ obtains the form:

$$\mathbb{R}(\zeta) = (1-\zeta)^{-1} \log \left( \sum_{q=0}^{\infty} (-1)^q \binom{\zeta(\varphi+1)+q}{q} (2\varphi)^\zeta \delta^{\zeta-1} \mathrm{B}\left(\zeta - \frac{\zeta}{\delta} + q + \frac{1}{\delta}, \zeta(\varphi-1)+1\right) \right).$$

The Havrda and Charvat entropy measure of the UEHLD is given by:

$$\hbar(\zeta) = \frac{1}{2^{1-\zeta}-1} \left[ \left( \int_0^1 (g(t))^\zeta dt \right)^{\frac{1}{\zeta}} - 1 \right], \zeta \neq 1, \zeta > 0$$

$$= \frac{1}{2^{1-\zeta}-1} \left[ \left( \sum_{q=0}^{\infty} (-1)^q \binom{\zeta(\varphi+1)+q}{q} (2\varphi)^\zeta \delta^{\zeta-1} \mathrm{B}\left(\zeta - \frac{\zeta}{\delta} + q + \frac{1}{\delta}, \zeta(\varphi-1)\right) \right)^{\frac{1}{\zeta}} - 1 \right].$$

The Tsallis entropy of the UEHLD is calculated as follows:

$$\aleph(\zeta) = \frac{1}{\zeta-1} \left[ 1 - \int_0^1 (g(t))^\zeta dt \right], \zeta \neq 1, \zeta > 0$$

$$= \frac{1}{\zeta-1} \left[ 1 - \left( \sum_{q=0}^{\infty} (-1)^q \binom{\zeta(\varphi+1)+q}{q} (2\varphi)^\zeta \delta^{\zeta-1} \mathrm{B}\left(\zeta - \frac{\zeta}{\delta} + q + \frac{1}{\delta}, \zeta(\varphi-1)\right) \right) \right].$$

Table 2 gives some numerical values of $\mathbb{R}(\zeta), \hbar(\zeta)$, and $\aleph(\zeta)$ for some selected parameter values.

**Table 2.** Entropy measures of the UEHLD.

| $\zeta$ | Measures | $\varphi = 0.5,$ $\delta = 0.5$ | $\varphi = 0.5,$ $\delta = 1$ | $\varphi = 0.5,$ $\delta = 1.5$ | $\varphi = 1,$ $\delta = 0.5$ | $\varphi = 1.5,$ $\delta = 0.5$ | $\varphi = 1,$ $\delta = 1$ | $\varphi = 1.5,$ $\delta = 1.5$ |
|---|---|---|---|---|---|---|---|---|
| | $\mathbb{R}(\zeta)$ | −0.229 | −0.098 | −0.185 | −0.56 | −1.022 | −0.064 | −0.099 |
| 0.8 | $\hbar(\zeta)$ | −0.374 | −0.162 | −0.303 | −0.878 | −1.516 | −0.106 | −0.165 |
| | $\aleph(\zeta)$ | −0.224 | −0.097 | −0.181 | −0.53 | −0.924 | −0.063 | −0.098 |
| | $\mathbb{R}(\zeta)$ | −0.682 | −0.365 | −0.565 | −1.557 | −2.295 | −0.118 | −0.154 |
| 1.5 | $\hbar(\zeta)$ | −0.872 | −0.441 | −0.708 | −2.323 | −3.924 | −0.137 | −0.18 |
| | $\aleph(\zeta)$ | −0.813 | −0.4 | −0.653 | −2.356 | −4.302 | −0.121 | −0.161 |

We conclude from Table 2 that, when the value of $\zeta$ increases, all entropy measures decrease, resulting in greater information. When the value of $\delta$ rises, for the same value of $\varphi$, we conclude that the $\mathbb{R}(\zeta), \hbar(\zeta)$, and $\aleph(\zeta)$ measures decrease, implying that there is less variability. Furthermore, we infer that the $\mathbb{R}(\zeta), \hbar(\zeta)$, and $\aleph(\zeta)$ measurements decrease as the value of $\varphi$ rises, for the same value of $\delta$, implying decreased variability.

## 5. Estimation of the UEHLD's Parameters

The parameter estimators of the UEHLD, using the ML, MPS and Bayesian methods, are discussed in this section. The approximate CI and credible Bayesian intervals are given.

### 5.1. ML Estimators

Assume $t_1, \ldots, t_n$ are the observed values from the UEHLD with parameters $\varphi$, and $\delta$. The likelihood function, say $L(\underline{t}|\varphi, \delta)$, of the UEHLD is expressed as:

$$L(\underline{t}|\varphi, \delta) = (2\varphi\delta)^n \prod_{i=1}^{n} \frac{t_i^{\delta-1}}{\left(1+t_i^\delta\right)^2} \left( \frac{1-t_i^\delta}{1+t_i^\delta} \right)^{\varphi-1}. \tag{14}$$

Then, the log likelihood function, say $\ell_1$, of the UEHLD is given as

$$\begin{aligned}
\ell_1 &= n[\ln(2) + \ln(\varphi) + \ln(\delta)] + (\delta - 1)\sum_{i=1}^{n}\ln(t_i) - 2\sum_{i=1}^{n}\ln\left(1 + t_i{}^{\delta}\right) \\
&\quad + (\varphi - 1)\sum_{i=1}^{n}\left[\ln\left(1 - t_i{}^{\delta}\right) - \ln\left(1 + t_i{}^{\delta}\right)\right].
\end{aligned}$$

Therefore, the ML equations are given by:

$$\frac{\partial \ell_1}{\partial \varphi} = \frac{n}{\varphi} + \sum_{i=1}^{n}\left[\ln\left(1 - t_i{}^{\delta}\right) - \ln\left(1 + t_i{}^{\delta}\right)\right],$$

$$\frac{\partial \ell_1}{\partial \delta} = \frac{n}{\delta} + \sum_{i=1}^{n}\ln(t_i) - 2\sum_{i=1}^{n}\frac{t_i{}^{\delta}\ln(t_i)}{1 + t_i{}^{\delta}} - (\varphi - 1)\sum_{i=1}^{n}\left[\frac{2t_i{}^{\delta}\ln(t_i)}{1 - t_i{}^{2\delta}}\right].$$

Solving the non-linear equations $\partial \ell_1/\partial \varphi = 0$, and $\partial \ell_1/\partial \delta$ numerically by using optimization algorithms such as the Newton–Raphson (NR) algorithm, we determine the ML estimators of $\varphi$, and $\delta$.

Furthermore, it is known that, under regularity conditions, the asymptotic distribution of the ML estimators of the UEHLD parameters is given by:

$$(\hat{\varphi} - \varphi), (\hat{\delta} - \delta) \sim N(0, I^{-1}(\varphi, \delta)),$$

where $I^{-1}(\varphi, \delta)$ is the variance–covariance matrix of the UEHLD parameters. Therefore, the two-sided approximate $(1 - \alpha)\%$ CIs for the ML estimates of $\varphi, \delta$ can be obtained as follows:

$$L_{\varphi} = \hat{\varphi} - z_{\alpha/2}\sqrt{\mathrm{var}(\hat{\varphi})}, U_{\varphi} = \hat{\varphi} + z_{\alpha/2}\sqrt{\mathrm{var}(\hat{\varphi})},$$

and

$$L_{\delta} = \hat{\delta} - z_{\alpha/2}\sqrt{\mathrm{var}(\hat{\delta})}, U_{\delta} = \hat{\delta} + z_{\alpha/2}\sqrt{\mathrm{var}(\hat{\delta})}.$$

where $z_{\alpha/2}$ is the $100(1 - \alpha)\%$ th standard normal percentile and var(.) denotes the diagonal elements of the variance–covariance matrix corresponding to the model parameters.

### 5.2. MPS Estimator

A strong alternative method known as MPS was introduced by Cheng and Amin [30] for determining the population parameters of continuous distributions. Take a look at the ordered products $T_{(1)}, T_{(2)}, \ldots, T_{(n)}$, which constitute a random sample of size $n$ drawn from CDF (2). Hence, the geometric mean $D^{\bullet}$ of the product spacing function is defined by:

$$D^{\bullet} = \left\{\prod_{i=1}^{n+1} P_i\right\}^{1/(n+1)},$$

$$P_i = \begin{cases} P_1 = F(t_1, \varphi, \delta) \\ P_i = F(t_i, \varphi, \delta) - F(t_{i-1}, \varphi, \delta) & i = 2, \ldots, n \\ P_{n+1} = 1 - F(t_n, \varphi, \delta) \end{cases}$$

such that $\sum P_i = 1$; for simplicity, we write $t_i$ instead of $t_{(i)}$. Then, the product spacing function is

$$D^{\bullet} = \left\{1 - \left(\frac{1 - t_1^{\delta}}{1 + t_1^{\delta}}\right)^{\varphi}\left(\frac{1 - t_n^{\delta}}{1 + t_n^{\delta}}\right)^{\varphi}\prod_{i=2}^{n+1}\left(\frac{1 - t_{i-1}^{\delta}}{1 + t_{i-1}^{\delta}}\right)^{\varphi} - \left(\frac{1 - t_i^{\delta}}{1 + t_i^{\delta}}\right)^{\varphi}\right\}^{1/(n+1)}.$$

The natural logarithm of the product spacing function is

$$\ln D^{\bullet} = \frac{1}{n+1}\left\{\ln\left[1 - \left(\frac{1 - t_1^{\delta}}{1 + t_1^{\delta}}\right)^{\varphi}\right] + \varphi\ln\left(\frac{1 - t_n^{\delta}}{1 + t_n^{\delta}}\right) + \sum_{i=2}^{n}\ln\left[\left(\frac{1 - t_{i-1}^{\delta}}{1 + t_{i-1}^{\delta}}\right)^{\varphi} - \left(\frac{1 - t_i^{\delta}}{1 + t_i^{\delta}}\right)^{\varphi}\right]\right\}. \tag{15}$$

Partially differentiate (15) from $\varphi$, and $\delta$, then equal them to zero. By using numerical analysis, it is possible to find the estimators $\hat{\varphi}, \hat{\delta}$ of $\varphi, \delta$ as the non-linear equations' solutions.

### 5.3. Bayesian Estimators

Here, we obtain the Bayesian estimator of the UEHLD parameters. The Bayesian estimators of $\varphi$, and $\delta$ are regarded under the squared error loss function (SELF), which are, respectively, defined by:

$$L(\widetilde{\varphi}, \varphi) = (\widetilde{\varphi} - \varphi)^2, L(\widetilde{\delta}, \delta) = \left(\widetilde{\delta} - \delta\right)^2.$$

Assume that the prior distribution of $\varphi, \delta$ denoted by $\pi(\varphi), \pi(\delta)$ has an independent gamma distribution. The joint gamma prior density of $\varphi, \delta$ can be written as

$$\pi(\varphi, \delta) \propto \varphi^{q_1-1} e^{-w_1\varphi} \delta^{q_2-1} e^{-w_2\delta}; q_j, w_j > 0, j = 1, 2. \tag{16}$$

The ML estimator for $\varphi$ and $\delta$ is obtained by equating the estimates and their variances with the inverse of the Fisher information matrix of $\varphi$ and $\delta$ in order to extract the hyperparameters of the informative priors (see Dey et al. [31] for more information). The joint posterior of the UEHLD with parameters $\varphi$ and $\delta$ is obtained using (14) and (16) as:

$$\pi(\varphi, \delta | \underline{t}) \propto \pi(\varphi, \delta) L(\underline{t} | \varphi, \delta).$$

Then, the joint posterior can be written as:

$$\pi(\varphi, \delta | \underline{t}) \propto \varphi^{q_1+n-1} e^{-w_1\varphi} \delta^{q_2+n-1} e^{-w_2\delta} \prod_{i=1}^{n} \frac{t_i^{\delta-1}}{\left(1+t_i^{\delta}\right)^2} \left(\frac{1-t_i^{\delta}}{1+t_i^{\delta}}\right)^{\varphi-1}.$$

We can employ the Markov Chain Monte Carlo (MCMC) method to acquire the Bayesian estimators. Gibbs sampling and the more general Metropolis within Gibbs samplers are useful subclasses of the MCMC techniques. The two most-well-known MCMC methods are the Gibbs sampling and Metropolis–Hastings (MH) algorithms. We created random samples from conditional posterior densities of $\varphi, \delta$ using the MH inside the Gibbs sampling steps as follows:

$$\pi(\varphi | \delta, \underline{t}) \propto \varphi^{q_1+n-1} e^{-\varphi[w_1 - \sum_{i=1}^{n} \ln\left(\frac{1-t_i^{\delta}}{1+t_i^{\delta}}\right)]} \sim \text{Gamma}\left(q_1+n, \left[w_1 - \sum_{i=1}^{n} \ln\left(\frac{1-t_i^{\delta}}{1+t_i^{\delta}}\right)\right]\right),$$

and

$$\pi(\delta | \varphi, \underline{t}) \propto \delta^{q_2+n-1} e^{-w_2\delta} \prod_{i=1}^{n} \frac{t_i^{\delta-1}}{\left(1+t_i^{\delta}\right)^2} \left(\frac{1-t_i^{\delta}}{1+t_i^{\delta}}\right)^{\varphi-1}.$$

The Bayesian estimators were obtained via SELF. The 95% two-sided highest density credible region interval for the unknown parameters or any function of them is given as $[\varphi_{0.025N:N}, \varphi_{0.975N:N}]$ and $[\delta_{0.025N:N}, \delta_{0.975N:N}]$ by using the method proposed by Chen and Shao [32].

## 6. Performance Analysis by Monte Carlo Simulation

In this section, a Monte Carlo simulation experiment is carried out to analyze the performance of point estimates in terms of the bias and MSE, as well as the performance of the interval estimates in terms of the CI length (L.CI). With various parameter values and sample sizes in mind, the simulation study was carried out. This section is broken into two sections, the first of which is a simulation study and the second of which outlines findings of the simulation.

*6.1. Simulation Study*

First, we set the true value with various parameter values of the UEHLD as: $\varphi = 4$ and $\delta = 0.5,\ 2,\ 4$ in Table 3, while $\varphi = 0.5,\ 2$ and $\delta = 0.5,\ 2,\ 4$ in Table 4. Altogether, nine sets of simulations of the UEHLD data with different sample sizes as $n = 30, 75$, and $150$ were generated. To avoid the starting bias, MSE, and length of CIs, 10,000 points were generated for each sample simulation. The generated data of the UEHLD were obtained by using QF (4). The estimates of the ML, MPS, and Bayesian techniques were obtained, and we used the NR algorithm for the numerical analysis to obtain the ML estimates (MLEs) and MPS estimates, as well as the MH algorithm to obtain the Bayesian estimates. The iterative algorithms were used to obtain 10,000 estimates for each parameter of the UEHLD when the first initial one was the actual parameter. In CI, we used the 5% level of significance. This simulation study was implemented via R packages.

Simulation algorithm: By creating all simulation controls, we may develop our model. The steps below must be completed in this stage in the following order:

Step 1: Assume various values for the sample size, as well as the UEHLD parameter vector.
Step 2: Using the QF, create the sample random values for the UEHLD.
Step 3: To acquire the estimators of the parameters for the UEHLD, we computed by solving non-linear equations for each estimate technique.
Step 4: Perform this experiment $(L - 1)$ times.

**Table 3.** Different estimates for the UEHLD parameters at $\varphi = 4$.

| | | | | ML | | | MPS | | | Bayesian | | |
|---|---|---|---|---|---|---|---|---|---|---|---|---|
| $\varphi$ | $\delta$ | $n$ | | Bias | MSE | LACI | Bias | MSE | LACI | Bias | MSE | LCCI |
| 4 | 0.5 | 30 | | 2.0073 | 5.2698 | 4.3686 | 1.1855 | 2.0800 | 3.2213 | 0.0099 | 0.0206 | 0.5464 |
| | | | $\delta$ | 0.2168 | 0.0690 | 0.5818 | 0.1290 | 0.0308 | 0.4674 | 0.0349 | 0.0074 | 0.2975 |
| | | 75 | $\varphi$ | 1.7336 | 3.3980 | 2.4577 | 1.4091 | 2.2988 | 2.1950 | 0.0082 | 0.0793 | 0.3429 |
| | | | $\delta$ | 0.1835 | 0.0402 | 0.3161 | 0.1474 | 0.0270 | 0.2856 | 0.0310 | 0.0342 | 0.1868 |
| | | 150 | $\varphi$ | 1.7281 | 3.2058 | 1.8377 | 1.0685 | 1.9471 | 1.6054 | 0.0078 | 0.0298 | 0.2079 |
| | | | $\delta$ | 0.1817 | 0.0358 | 0.2078 | 0.1180 | 0.0312 | 0.1958 | 0.0235 | 0.0169 | 0.1289 |
| | 2 | 30 | $\varphi$ | 1.3661 | 2.3880 | 2.8328 | 0.8318 | 1.0361 | 2.3012 | 0.0060 | 0.0204 | 0.5552 |
| | | | $\delta$ | 1.7326 | 4.2751 | 4.4255 | 0.9446 | 1.3462 | 2.6423 | 0.0116 | 0.0187 | 0.5409 |
| | | 75 | $\varphi$ | 1.2307 | 1.7054 | 1.7132 | 0.9823 | 1.1135 | 1.5118 | 0.0042 | 0.0740 | 0.3301 |
| | | | $\delta$ | 1.3928 | 2.2318 | 2.1193 | 1.0534 | 1.2833 | 1.6344 | 0.0124 | 0.0796 | 0.3451 |
| | | 150 | $\varphi$ | 1.2105 | 1.5649 | 1.2379 | 0.7713 | 1.0360 | 1.1219 | 0.0044 | 0.0290 | 0.2062 |
| | | | $\delta$ | 1.3343 | 1.9335 | 1.5348 | 0.8278 | 1.2356 | 1.2745 | 0.0075 | 0.0314 | 0.2189 |
| | 4 | 30 | $\varphi$ | 1.2142 | 1.8066 | 2.2608 | 0.5266 | 0.4207 | 1.4846 | 0.0025 | 0.0191 | 0.5404 |
| | | | $\delta$ | 2.3127 | 8.8163 | 8.9583 | 1.4443 | 2.6906 | 3.0498 | 0.0076 | 0.0211 | 0.5526 |
| | | 75 | $\varphi$ | 1.1564 | 1.4955 | 1.5608 | 0.6366 | 0.4640 | 0.9504 | 0.0051 | 0.0770 | 0.3365 |
| | | | $\delta$ | 2.0123 | 7.7098 | 6.3376 | 1.8318 | 3.6308 | 2.0584 | 0.0075 | 0.0837 | 0.3514 |
| | | 150 | $\varphi$ | 1.1326 | 1.3535 | 1.0424 | 0.4978 | 0.4368 | 0.6574 | 0.0012 | 0.0283 | 0.2052 |
| | | | $\delta$ | 1.7292 | 6.9873 | 3.8550 | 1.3834 | 2.7085 | 1.3902 | 0.0032 | 0.0321 | 0.2203 |

**Table 4.** Different estimates for the UEHLD parameters at $\varphi = 0.5, 2$.

| $\varphi$ | $\delta$ | $n$ | | **ML** Bias | MSE | LACI | **MPS** Bias | MSE | LACI | **Bayesian** Bias | MSE | LCCI |
|---|---|---|---|---|---|---|---|---|---|---|---|---|
| 0.5 | 0.5 | 30 | $\varphi$ | 0.2473 | 0.0782 | 0.5112 | 0.1569 | 0.0369 | 0.4355 | 0.0449 | 0.0089 | 0.3123 |
| | | | $\delta$ | 0.2129 | 0.0659 | 0.5623 | 0.1312 | 0.0313 | 0.4656 | 0.0353 | 0.0074 | 0.2889 |
| | | 75 | $\varphi$ | 0.2269 | 0.0587 | 0.3327 | 0.1894 | 0.0421 | 0.3087 | 0.0418 | 0.0433 | 0.1996 |
| | | | $\delta$ | 0.1886 | 0.0417 | 0.3070 | 0.1549 | 0.0292 | 0.2823 | 0.0328 | 0.0396 | 0.2008 |
| | | 150 | $\varphi$ | 0.2153 | 0.0498 | 0.2317 | 0.1551 | 0.0411 | 0.2216 | 0.0310 | 0.0237 | 0.1468 |
| | | | $\delta$ | 0.1793 | 0.0352 | 0.2164 | 0.1277 | 0.0286 | 0.2063 | 0.0265 | 0.0189 | 0.1348 |
| | 2 | 30 | $\varphi$ | 0.1395 | 0.0289 | 0.2991 | 0.0921 | 0.0128 | 0.2565 | 0.0242 | 0.0030 | 0.1868 |
| | | | $\delta$ | 1.2332 | 2.1865 | 2.5161 | 0.7204 | 0.7119 | 1.7227 | 0.0080 | 0.0198 | 0.5589 |
| | | 75 | $\varphi$ | 0.1380 | 0.0222 | 0.2216 | 0.1098 | 0.0139 | 0.1700 | 0.0198 | 0.0130 | 0.1174 |
| | | | $\delta$ | 1.1628 | 1.5706 | 1.8335 | 0.8738 | 0.8563 | 1.1949 | 0.0148 | 0.0739 | 0.3342 |
| | | 150 | $\varphi$ | 0.1397 | 0.0207 | 0.1352 | 0.0883 | 0.0129 | 0.1222 | 0.0147 | 0.0068 | 0.0847 |
| | | | $\delta$ | 1.2083 | 1.5556 | 1.2131 | 0.6919 | 0.7262 | 0.8620 | 0.0121 | 0.0302 | 0.2152 |
| | 4 | 30 | $\varphi$ | 0.0646 | 0.0062 | 0.1777 | 0.0404 | 0.0031 | 0.1507 | 0.0140 | 0.0013 | 0.1283 |
| | | | $\delta$ | 1.2262 | 1.1192 | 3.0775 | 0.6686 | 0.5838 | 1.4502 | 0.0025 | 0.0221 | 0.5722 |
| | | 75 | $\varphi$ | 0.0459 | 0.0044 | 0.1174 | 0.0462 | 0.0027 | 0.0933 | 0.0104 | 0.0026 | 0.0828 |
| | | | $\delta$ | 0.8027 | 0.9243 | 2.0755 | 0.8254 | 0.7335 | 0.8957 | 0.0077 | 0.0777 | 0.3443 |
| | | 150 | $\varphi$ | 0.0606 | 0.0042 | 0.0918 | 0.0324 | 0.0026 | 0.0672 | 0.0072 | 0.0029 | 0.0607 |
| | | | $\delta$ | 1.2160 | 0.8693 | 1.8136 | 0.6087 | 0.5203 | 0.6822 | 0.0068 | 0.0282 | 0.2026 |
| 2 | 0.5 | 30 | $\varphi$ | 0.9910 | 1.3135 | 2.2578 | 0.6300 | 0.6371 | 1.9222 | 0.0105 | 0.0185 | 0.5236 |
| | | | $\delta$ | 0.2071 | 0.0607 | 0.5225 | 0.1267 | 0.0283 | 0.4343 | 0.0350 | 0.0075 | 0.3021 |
| | | 75 | $\varphi$ | 0.8987 | 0.9215 | 1.3228 | 0.7403 | 0.6448 | 1.2201 | 0.0152 | 0.0775 | 0.3304 |
| | | | $\delta$ | 0.1932 | 0.0445 | 0.3330 | 0.1571 | 0.0307 | 0.3042 | 0.0309 | 0.0333 | 0.1887 |
| | | 150 | $\varphi$ | 0.8709 | 0.8106 | 0.8952 | 0.7882 | 0.6690 | 0.8565 | 0.0119 | 0.0270 | 0.1943 |
| | | | $\delta$ | 0.1820 | 0.0361 | 0.2144 | 0.1634 | 0.0294 | 0.2042 | 0.0235 | 0.0168 | 0.1243 |
| | 2 | 30 | $\varphi$ | 0.6791 | 0.6031 | 1.4772 | 0.4232 | 0.2815 | 1.2551 | 0.0085 | 0.0166 | 0.4915 |
| | | | $\delta$ | 1.6394 | 3.7521 | 4.0461 | 0.9140 | 1.2861 | 2.6331 | 0.0112 | 0.0197 | 0.5318 |
| | | 75 | $\varphi$ | 0.6189 | 0.4290 | 0.8414 | 0.5037 | 0.2921 | 0.7682 | 0.0155 | 0.0619 | 0.2932 |
| | | | $\delta$ | 1.3982 | 2.2541 | 2.1450 | 1.0811 | 1.3608 | 1.7192 | 0.0132 | 0.0788 | 0.3385 |
| | | 150 | $\varphi$ | 0.5957 | 0.3789 | 0.6091 | 0.4000 | 0.2689 | 0.5739 | 0.0096 | 0.0246 | 0.1892 |
| | | | $\delta$ | 1.3239 | 1.8850 | 1.4259 | 0.8927 | 1.3390 | 1.2459 | 0.0066 | 0.0301 | 0.2133 |
| | 4 | 30 | $\varphi$ | 0.5359 | 0.3686 | 1.1194 | 0.2775 | 0.1194 | 0.8075 | 0.0141 | 0.0148 | 0.4686 |
| | | | $\delta$ | 1.5037 | 15.2652 | 6.7812 | 1.5006 | 2.9003 | 3.1584 | 0.0082 | 0.0201 | 0.5378 |
| | | 75 | $\varphi$ | 0.5223 | 0.2997 | 0.6432 | 0.3178 | 0.1165 | 0.4886 | 0.0112 | 0.0556 | 0.2958 |
| | | | $\delta$ | 1.4833 | 6.3802 | 4.3790 | 1.8346 | 3.6374 | 2.0437 | 0.0117 | 0.0842 | 0.3511 |
| | | 150 | $\varphi$ | 0.4947 | 0.2627 | 0.5260 | 0.2535 | 0.1141 | 0.3576 | 0.0062 | 0.0220 | 0.1774 |
| | | | $\delta$ | 0.9451 | 2.2699 | 3.2671 | 1.3503 | 2.6338 | 1.4620 | 0.0064 | 0.0312 | 0.2140 |

*6.2. Simulation Results*

The outcomes of the proposed methods for estimating the point and interval parameters are displayed in Tables 3 and 4. They provide the results and some interesting information. The following comments can be made:

■ As the sample size increases, the estimates become increasingly precise, indicating that they are asymptotically unbiased.

■ When the MSE value is close to zero, the parameter estimates are from the best unbiased estimator.

■ The MSE decreases in each estimate as the sample size increases, indicating consistency among the different estimates.

■ At true value $\varphi = 0.5$ and as the value of $\delta$ increases from 2 to 4, the MSE of both estimates, based on three different techniques, decreases.

■ As the true value of $\varphi$ increases, the MSE of both estimates, based on three different techniques, increases at the same true value of $\delta$.

■ The MSE and length of the CI for the MPS estimates is smaller than the ML estimates for all true parameter values.

■ Both parameter estimates have the largest MSE for the three proposed methods at the true value of $\varphi = \delta = 4$, except a few cases.

■ In the majority of situations, we conclude that the MPS estimates are preferable compared to the ML estimates due to their precision measures being the smallest.

■ As $n$ grows larger, the length of the CI (ML: LACI; MPS: LACI; Bayesian: LCCI) for the estimates decreases, suggesting that the CI is the shortest.

■ For all true parameter values, the LACI of the MPS estimate is lower than the LACI of the ML estimates.

■ The length of the CI for both estimates obtains its largest value, based on the three suggested methods, as the true values of the parameters increase.

## 7. Real Data Applications

We used the traditional value of criteria (VC) to compare the fit models, such as the Akaike information criterion (AIVC), consistent AIVC (CAIVC), Bayesian information criterion (BIVC), Hannan–Quinn information criterion (HQIVC), Anderson–Darling value (ADV), Cramer–von Mises value (CMV), Kolmogorov–Smirnov distance (KSD), *p*-value of Kolmogorov–Smirnov (PKS), and standard error (SE). Our primary statistical goal was to use a fitting approach model to examine three real datasets that are significant in different fields. In this respect, we compared the fit of the proposed UEHLD with that of the unit Weibull (UW), the Kumaraswamy (K), beta (Beta), Kumaraswamy–Kumaraswamy (KK) (El-Sherpieny and Ahmed [33]), Marshall–Olkin–Kumaraswamy (MOK) (George and Thobias [34]), UBXII, and UGLBXII distributions.

The effectiveness of the parameter estimator for the UEHLD for the three datasets under consideration was also assessed using the ML, MPS, and Bayesian techniques via the standard error and confidence interval length criteria measures. We obtained the estimators of the new model for three techniques for the datasets under consideration, with the exception of the first dataset, for which the MPS approach was not employed because it has more equal values. For further clarification, the log-likelihood of the suggested model is supplied, along with examples of the contour plots with various parameter values. We also provide plots of the posterior distributions of the parameters, as well as histograms for the marginal posterior density estimates for three datasets.

**Dataset I**: The trade share dataset takes into account the values of the trade share variable used in the renowned "Determinants of Economic Growth Data". Along with factors that may be associated with growth, the growth rates of up to 61 different countries were taken into consideration. The information is publicly accessible online as an addition to Stock and Watson [35]. The trade share dataset consists of the following numbers: 0.1405, 0.1566, 0.1577, 0.1604, 0.1608, 0.2215, 0.2994, 0.3131, 0.3246, 0.3247, 0.3295, 0.3300, 0.3379, 0.3397, 0.3523, 0.3589, 0.3933, 0.4176, 0.4258, 0.4356, 0.4421, 0.4444, 0.4505, 0.4558, 0.4683, 0.4733, 0.4846, 0.4889, 0.5096, 0.5177, 0.5278, 0.5347, 0.5433, 0.5442, 0.5508, 0.5527, 0.5606, 0.5607, 0.5671, 0.5753, 0.5828, 0.6030, 0.6050, 0.6136, 0.6261, 0.6395, 0.6469, 0.6512, 0.6816, 0.6994, 0.7048, 0.7292, 0.7430, 0.7455, 0.7798, 0.7984, 0.8147, 0.8230, 0.8302, 0.8342, 0.9794.

**Dataset II**: This dataset includes 30 measurements of polyester fibers' tensile strength made by Quesenberry and Hales [36]. The data are 0.023, 0.032, 0.054, 0.069, 0.081, 0.094, 0.105, 0.127, 0.148, 0.169, 0.188, 0.216, 0.255, 0.277, 0.311, 0.361, 0.376, 0.395, 0.432, 0.463, 0.481, 0.519, 0.529, 0.567, 0.642, 0.674, 0.752, 0.823, 0.887, 0.926.

**Dataset III**: COVID-19 of Britain: This dataset covered a period of 82 days, from 1 May 2021 to 16 July 2021 (see Abu El Azm et al. [37]). The following information is created using daily new deaths (DNDs), daily cumulative cases (DCCs), and daily cumulative deaths (DCDs): 0.0023, 0.0023, 0.0023, 0.0046, 0.0065, 0.0067, 0.0069, 0.0069, 0.0091, 0.0093, 0.0093, 0.0093, 0.0111, 0.0115, 0.0116, 0.0116, 0.0119, 0.0133, 0.0136, 0.0138, 0.0138, 0.0159, 0.0161, 0.0162, 0.0162, 0.0162, 0.0163, 0.0180, 0.0187, 0.0202, 0.0207, 0.0208, 0.0225, 0.0230, 0.0230, 0.0239, 0.0245, 0.0251, 0.0255, 0.0255, 0.0271, 0.0275, 0.0295, 0.0297, 0.0300, 0.0302, 0.0312, 0.0314, 0.0326, 0.0346, 0.0349, 0.0350, 0.0355, 0.0379, 0.0384, 0.0394, 0.0394, 0.0412, 0.0419, 0.0425, 0.0461, 0.0464, 0.0468, 0.0471, 0.0495, 0.0501, 0.0521, 0.0571, 0.0588, 0.0597, 0.0628, 0.0679, 0.0685, 0.0715, 0.0766, 0.0780, 0.0942, 0.0960, 0.0988, 0.1223, 0.1343, and 0.1781.

$$x_i = \left( \frac{DND_i}{DCC_i - DCD_{i-1}} \right) \times 1000.$$

Some descriptive statistics for the proposed datasets are displayed in Table 5 and represented in Figure 4.

**Table 5.** Descriptive summary datasets.

| Dataset | Min | Q(0.25) | Q(0.5) | Mean | Q(0.75) | Max |
|---------|--------|---------|--------|---------|---------|--------|
| I | 0.0110 | 0.1410 | 0.1510 | 0.2434 | 0.326 | 0.9490 |
| II | 0.0230 | 0.1323 | 0.3360 | 0.3659 | 0.5265 | 0.9260 |
| III | 0.0023 | 0.01432 | 0.0273 | 0.03571 | 0.04632 | 0.1781 |

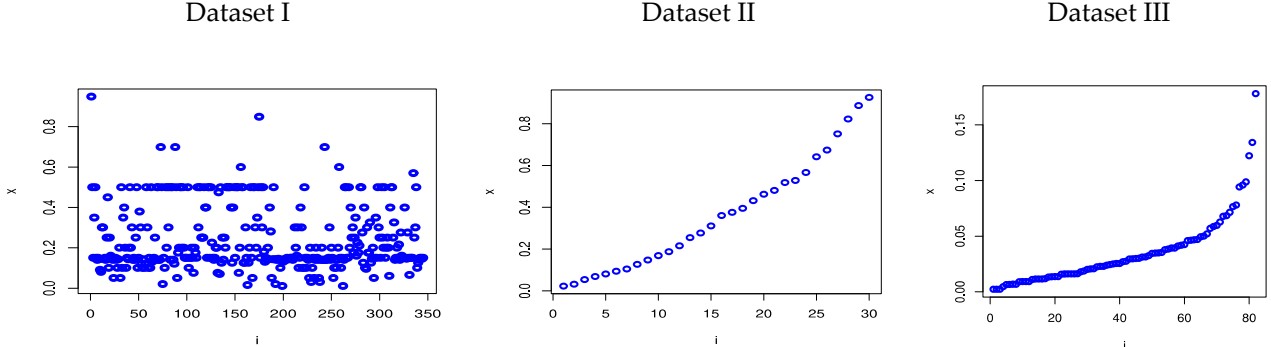

**Figure 4.** Data description.

The analysis of the three real datasets is discussed in more detail in the following subsections.

### 7.1. Analysis of First Dataset

**First**, in order to compare the fit models, we employed the aforementioned criterion measurements included in Table 6. Figure 5 displays the dataset's P-P plots, the fit UEHLD PDF plots with their empirical CDF, and the relative histogram with the fit UEHLD. These graphical goodness-of-fit methods in Figure 5 also corroborate the results in Table 6.

**Table 6.** MLE, SE, and measures of models for trade share data.

|  |  | Estimates | SE | KSD | PVK | AIVC | BIVC | CAIVC | HQIVC | CMV | ADV |
|---|---|---|---|---|---|---|---|---|---|---|---|
| UEHLD | $\varphi$ | 2.6743 | 0.3089 | 0.0519 | 0.9937 | −25.0405 | −20.8187 | −24.8336 | −23.3859 | 0.0391 | 0.3333 |
|  | $\delta$ | 2.0209 | 0.3791 |  |  |  |  |  |  |  |  |
| UW | $\alpha$ | 1.3395 | 0.1725 | 0.0682 | 0.9208 | −24.4872 | −20.2654 | −24.2803 | −22.8326 | 0.0630 | 0.5097 |
|  | $\beta$ | 1.7346 | 0.1695 |  |  |  |  |  |  |  |  |
| K | a | 2.3297 | 0.3055 | 0.0690 | 0.9141 | −23.2503 | −19.0285 | −23.0434 | −21.5957 | 0.0527 | 0.4005 |
|  | b | 2.7629 | 0.5550 |  |  |  |  |  |  |  |  |
| Beta | $\alpha$ | 2.7944 | 0.4881 | 0.0618 | 0.9629 | −23.9121 | −19.6903 | −23.7052 | −22.2576 | 0.0491 | 0.3864 |
|  | $\beta$ | 2.6041 | 0.4519 |  |  |  |  |  |  |  |  |
| KK | a | 4.6765 | 11.1153 | 0.0561 | 0.9850 | −20.7855 | −12.3420 | −20.0712 | −17.4764 | 0.0484 | 0.4016 |
|  | b | 0.8085 | 1.6378 |  |  |  |  |  |  |  |  |
|  | $\alpha$ | 2.4986 | 3.7154 |  |  |  |  |  |  |  |  |
|  | $\beta$ | 0.8259 | 1.2331 |  |  |  |  |  |  |  |  |
| MOK | $\alpha$ | 0.3008 | 0.3023 | 0.0582 | 0.9783 | −22.6367 | −16.3040 | −22.2156 | −20.1549 | 0.0490 | 0.4139 |
|  | $\beta$ | 3.0589 | 0.6447 |  |  |  |  |  |  |  |  |
|  | $\theta$ | 1.9501 | 0.9516 |  |  |  |  |  |  |  |  |
| UG | $\beta$ | 0.6161 | 0.2661 | 0.1098 | 0.4234 | −17.7518 | −13.5300 | −17.5449 | −16.0972 | 0.1585 | 1.1540 |
|  | $\theta$ | 1.0922 | 0.2472 |  |  |  |  |  |  |  |  |
| UGLBXII | $\alpha$ | 5.1555 | 8.0816 | 0.0548 | 0.9884 | −22.9406 | −16.6080 | −22.5196 | −20.4588 | 0.0411 | 0.3522 |
|  | $\beta$ | 0.9724 | 0.1862 |  |  |  |  |  |  |  |  |
|  | $\lambda$ | 1.8164 | 1.9159 |  |  |  |  |  |  |  |  |

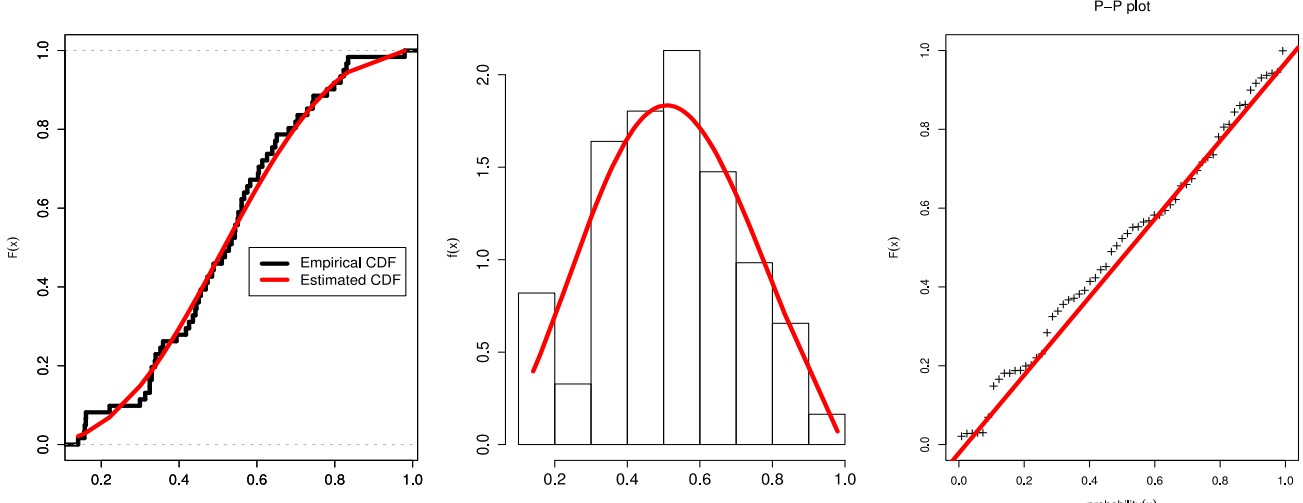

**Figure 5.** Plots of the estimated PDFs, CDFs, and P-P of the UEHLD of trade share data.

**Second**, from Table 5 and Figure 4, we cannot use the MPS method to estimate the parameters of Dataset I because this dataset has more equal values, then $F(t_i, \varphi, \delta) - F(t_{i-1}, \varphi, \delta)$ is equal to zero at most observations. Consequently, Table 7 only contains the ML and Bayesian estimates with the SELF of the UEHLD's parameters. 10,000 MCMC samples were generated using the MCMC method. To apply the MCMC sampler process, the starting values of the unknown parameters were assumed to represent their MLEs.

**Table 7.** Estimates with SE and CIs for UEHLD parameters for trade share data.

| | ML | | | | Bayesian | | | |
|---|---|---|---|---|---|---|---|---|
| | **Estimates** | **SE** | **Lower** | **Upper** | **Estimates** | **SE** | **Lower** | **Upper** |
| $\varphi$ | 2.6743 | 0.3089 | 1.4237 | 1.6860 | 1.5571 | 0.0587 | 1.4469 | 1.6742 |
| $\delta$ | 2.0209 | 0.3791 | 3.0081 | 4.2673 | 3.6527 | 0.2886 | 3.0643 | 4.1787 |

Figure 6 sketches the profile log-likelihood of the UEHLD for each parameter by fixing one parameter and varying the other. The figures show that the trade share dataset behaves very well, as we can see that the two roots of the parameters are global maxima. Figure 7 gives the contour plot with varying parameters and log-likelihoods of the UEHLD to confirm the estimates have unique points.

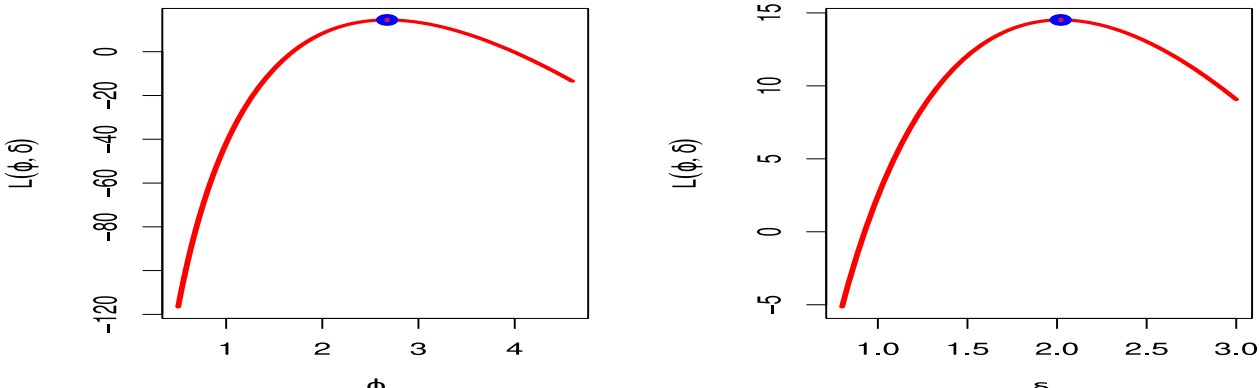

**Figure 6.** Profile likelihood for parameters of the UEHLD by trade share data.

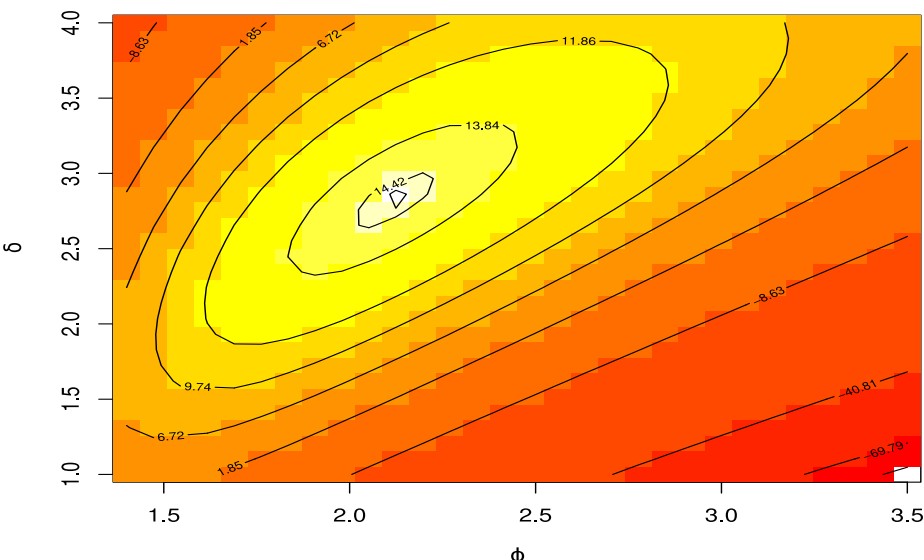

**Figure 7.** Contour plot of log-likelihood function with different UEHLD parameter values by trade share data.

Figure 8 shows the trace plots of the posterior distributions of the parameters to track the convergence of the MCMC outputs. This figure shows how well the MCMC process converges. Furthermore, this shows the histograms for the marginal posterior density estimates of the parameters based on 10,000 chain values and the Gaussian kernel. The estimations clearly show that all of the generated posteriors are symmetric with respect to the theoretical posterior density functions.

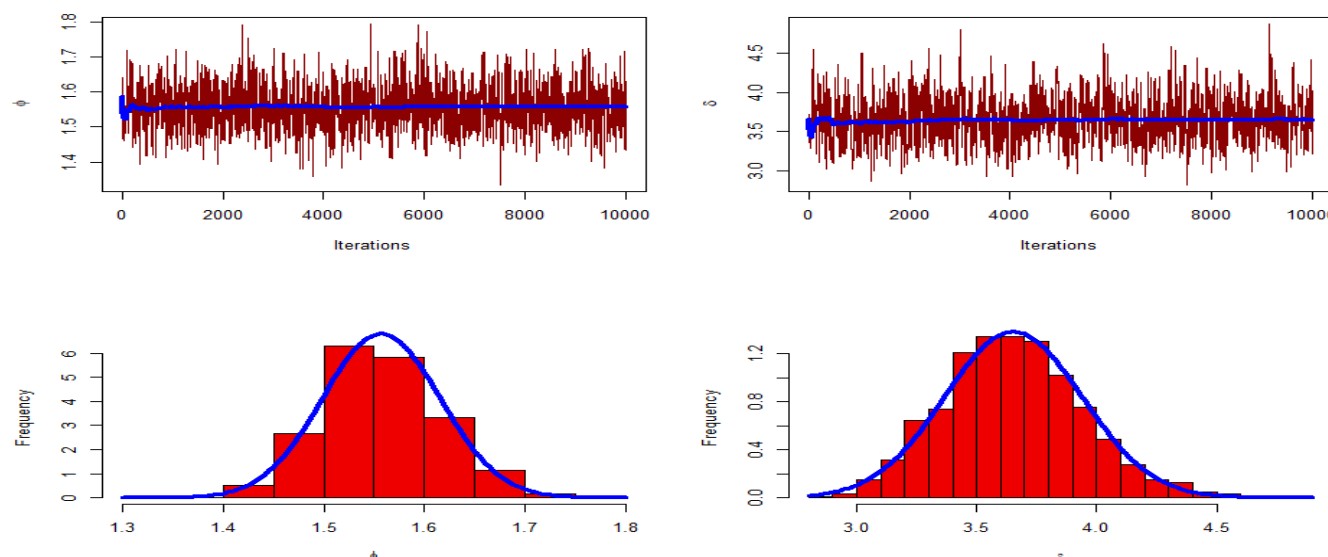

**Figure 8.** MCMC trace and posterior distribution for the UEHLD parameters by trade share data.

*7.2. Analysis of Second Dataset*

    **First**, we compared the fit of the proposed UEHLD with that of the beta, K, KK, MOK, UB, UBXII, and UGLBXII distributions. We used the aforementioned criterion measurements found in Table 8 to compare the fit models. Table 8 reveals that the UEHLD has smaller measures than the other competing distributions, indicating that it offers a superior fit.

**Table 8.** MLE, SE, and measures of models for polyester fibers' tensile strength data.

| | | Estimates | SE | KSD | PKS | AIVC | BIVC | CAIVC | HQIVC | CMV | ADV |
|---|---|---|---|---|---|---|---|---|---|---|---|
| UEHLD | $\varphi$ | 1.1281 | 0.2051 | 0.0565 | 0.9999 | −3.1043 | −0.3019 | −2.6599 | −2.2078 | 0.0150 | 0.1158 |
| | $\delta$ | 1.2346 | 0.2951 | | | | | | | | |
| K | a | 0.9627 | 0.2017 | 0.0650 | 0.9987 | −2.6221 | 0.1803 | −2.1776 | −1.7256 | 0.0183 | 0.1551 |
| | b | 1.6084 | 0.4137 | | | | | | | | |
| Beta | $\alpha$ | 0.9666 | 0.2238 | 0.0669 | 0.9979 | −2.6101 | 0.1923 | −2.1657 | −1.7136 | 0.0184 | 0.1559 |
| | $\beta$ | 1.6205 | 0.4107 | | | | | | | | |
| KK | a | 7.9330 | 1.2565 | 0.0714 | 0.9951 | −0.2150 | 5.3898 | 1.3850 | 1.5780 | 0.0163 | 0.1289 |
| | b | 0.4949 | 0.0478 | | | | | | | | |
| | $\alpha$ | 8.7493 | 0.0614 | | | | | | | | |
| | $\beta$ | 0.1404 | 0.0287 | | | | | | | | |
| MOK | $\alpha$ | 0.4365 | 0.4732 | 0.0628 | 0.9992 | −1.2087 | 2.9949 | −0.2856 | 0.1361 | 0.0151 | 0.1164 |
| | $\beta$ | 1.1872 | 0.3472 | | | | | | | | |
| | $\theta$ | 1.2585 | 0.6458 | | | | | | | | |
| UBXII | $\beta$ | 1.0331 | 0.2060 | 0.0993 | 0.9008 | 1.9220 | 4.7244 | 2.3665 | 2.8185 | 0.0586 | 0.4419 |
| | $\theta$ | 1.8465 | 0.3054 | | | | | | | | |
| UGLBXII | $\alpha$ | 582.6661 | 45.8072 | 0.0570 | 0.9999 | −1.4258 | 2.7778 | −0.5027 | −0.0811 | 0.0163 | 0.1166 |
| | $\beta$ | 0.6820 | 0.1076 | | | | | | | | |
| | $\lambda$ | 160.0645 | 25.8435 | | | | | | | | |

    The estimated PDF, empirical CDF, and P-P plots for Dataset II are shown in Figure 9. These graphical plots support the outcomes in Table 9.

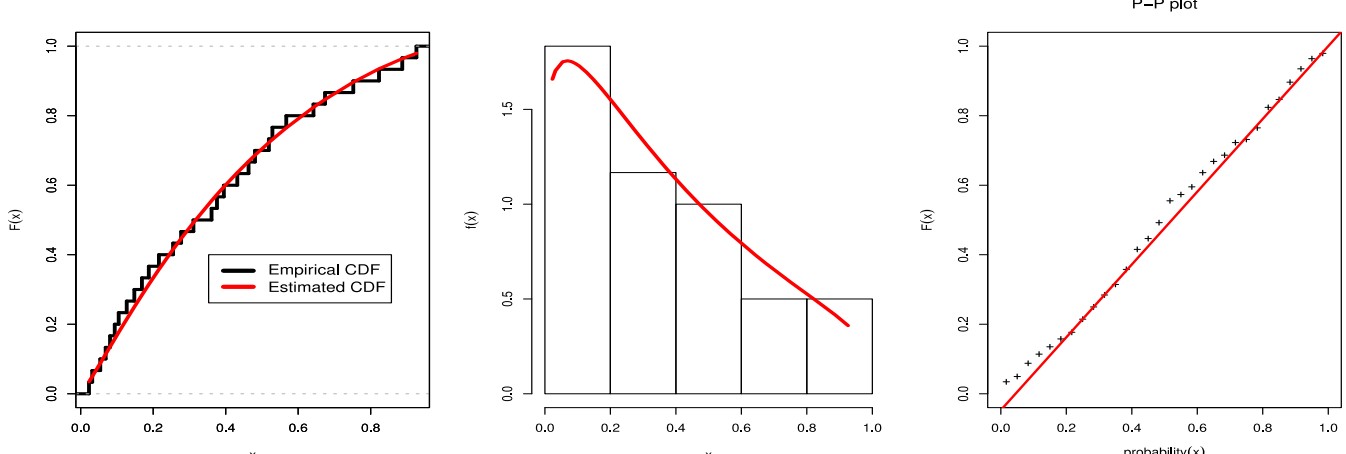

**Figure 9.** Plots of the estimated PDFs, CDFs, and P-P of the UEHLD of polyester fibers' tensile strength data.

**Table 9.** Estimates, SE, and CI of the UEHLD parameters for polyester fibers' tensile strength data.

| | ML | | | | MPS | | | | Bayesian | | | |
|---|---|---|---|---|---|---|---|---|---|---|---|---|
| | Estimates | SE | Lower | Upper | Estimates | SE | Lower | Upper | Estimates | SE | Lower | Upper |
| $\varphi$ | 1.1281 | 0.2051 | 0.7262 | 1.5301 | 1.0058 | 0.2074 | 0.5994 | 1.4124 | 1.1354 | 0.1931 | 0.7826 | 1.5369 |
| $\delta$ | 1.2346 | 0.2951 | 0.6563 | 1.8130 | 1.0606 | 0.2407 | 0.5889 | 1.5325 | 1.2423 | 0.2932 | 0.7414 | 1.8473 |

**Second**, utilizing the information on the tensile strength of the fibers, we determined the ML, MPS, and Bayesian estimates using the SELF of the UEHLD's parameters, which are listed in Table 9. We used the mentioned MCMC algorithm to generate 10,000 MCMC samples. The initial values of the unknown parameters were taken to be their MLEs in order to use the MCMC sampling procedure.

For the data on the tensile strength of polyester fibers, Figure 10 draws the profile log-likelihood of the UEHLD for each parameter by fixing one parameter and changing the others. This figure demonstrates the excellent behavior of the aforementioned data, since the two roots of the parameters are global maxima.

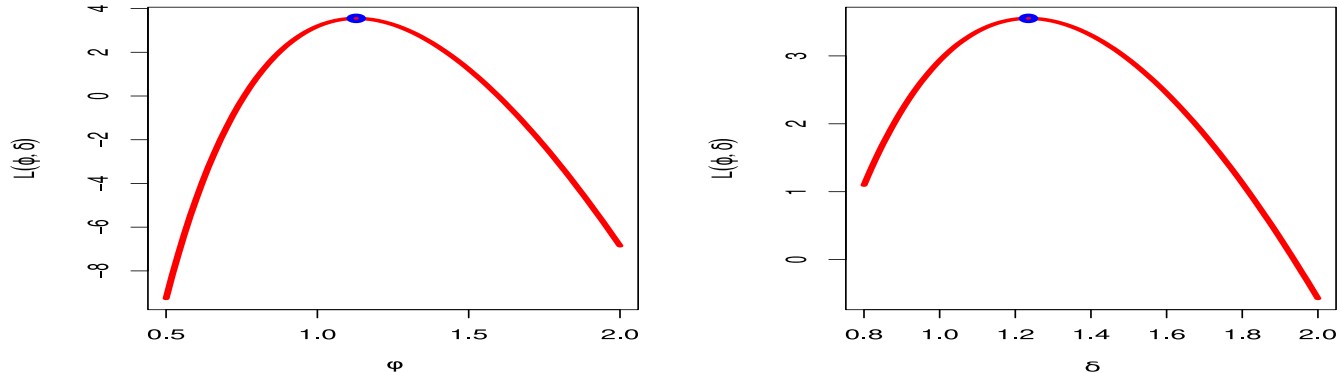

**Figure 10.** Profile likelihood for parameters of the UEHLD for polyester fibers' tensile strength data.

Figure 11 shows a contour plots with variable parameters, the log-likelihood function, and the log-product spacing function of the UEHLD to verify that the estimates have unique points.

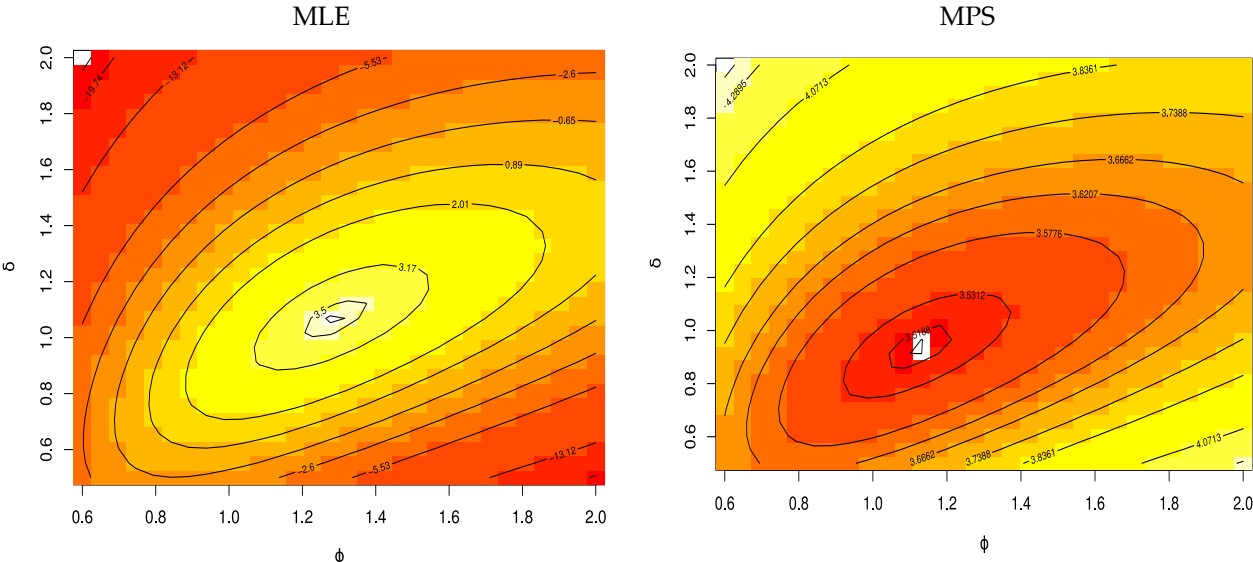

**Figure 11.** Contour plots of log functions with different UEHLD parameter values of polyester fibers' tensile data.

Figure 12 shows trace plots of the posterior distributions of the parameters to track the convergence of the MCMC outputs. Additionally, they display the histograms for the marginal posterior density estimates of the parameters based on 10,000 chain values and the Gaussian kernel, demonstrating how effectively the MCMC process converges. All of the produced posteriors are symmetric with regard to the theoretical posterior density functions, as shown by the estimations.

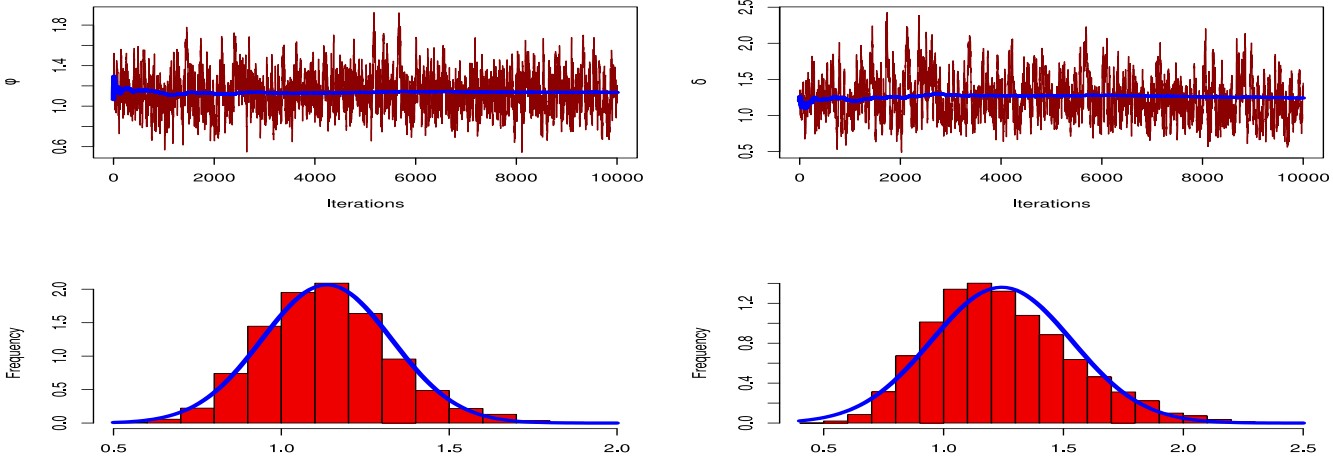

**Figure 12.** MCMC trace and posterior distribution for the UEHLD parameters using polyester fibers' tensile data.

### 7.3. Analysis of COVID-19 Data

**First**, we compared the fit of the proposed UEHLD with that of the UW, K, KK, MOK, and UG distributions. We used the aforementioned criterion measurements found in Table 10 to compare the fit models. Table 10 reveals that the UEHLD has smaller measures than other competing distributions, indicating that it offers a superior fit for COVID-19 data. We should also point out that, in comparison to the models reported by Abu El Azm et al. [37], the findings of our new model yield superior measure values.

**Table 10.** MLE, SE, and measures of models for COVID-19 data.

| | | Estimates | SE | KSD | PVKS | AICVC | BICVC | CAICVC | HQICVC | CVMV | ADV |
|---|---|---|---|---|---|---|---|---|---|---|---|
| UEHLD | $\varphi$ | 1.2515 | 0.1030 | 0.0574 | 0.9496 | −385.0542 | −380.2408 | −384.9023 | −383.1217 | 0.0555 | 0.3931 |
| | $\delta$ | 29.3838 | 9.3197 | | | | | | | | |
| UW | $\alpha$ | 0.0024 | 0.0003 | 0.0734 | 0.7695 | −381.6037 | −376.7903 | −381.4518 | −379.6712 | 0.0988 | 0.7075 |
| | $\beta$ | 4.3135 | 0.1105 | | | | | | | | |
| K | a | 1.2399 | 0.1055 | 0.0597 | 0.9322 | −384.6698 | −379.8564 | −384.5179 | −382.7373 | 0.0601 | 0.4228 |
| | b | 55.7476 | 18.3042 | | | | | | | | |
| KK | a | 2.4673 | 0.8418 | 0.0798 | 0.6729 | −379.0265 | −369.3996 | −378.5070 | −375.1614 | 0.0624 | 0.4916 |
| | b | 0.5254 | 0.1184 | | | | | | | | |
| | $\alpha$ | 3.9566 | 0.9116 | | | | | | | | |
| | $\beta$ | 3.8605 | 1.1342 | | | | | | | | |
| MOK | $\alpha$ | 0.0119 | 0.0225 | 0.1045 | 0.3324 | −372.2227 | −365.0025 | −371.9150 | −369.3239 | 0.0635 | 0.5250 |
| | $\beta$ | 1.3908 | 0.1793 | | | | | | | | |
| | $\theta$ | 1.8396 | 2.6201 | | | | | | | | |
| UG | $\beta$ | 0.0180 | 0.0071 | 0.1079 | 0.2953 | −363.6987 | −358.8852 | −363.5468 | −361.7661 | 0.2945 | 1.9852 |
| | $\theta$ | 0.9767 | 0.0803 | | | | | | | | |

The estimated PDF, empirical CDF, and P-P plots for the COVID-19 data are shown in Figure 13. These graphical plots support the outcomes in Table 10.

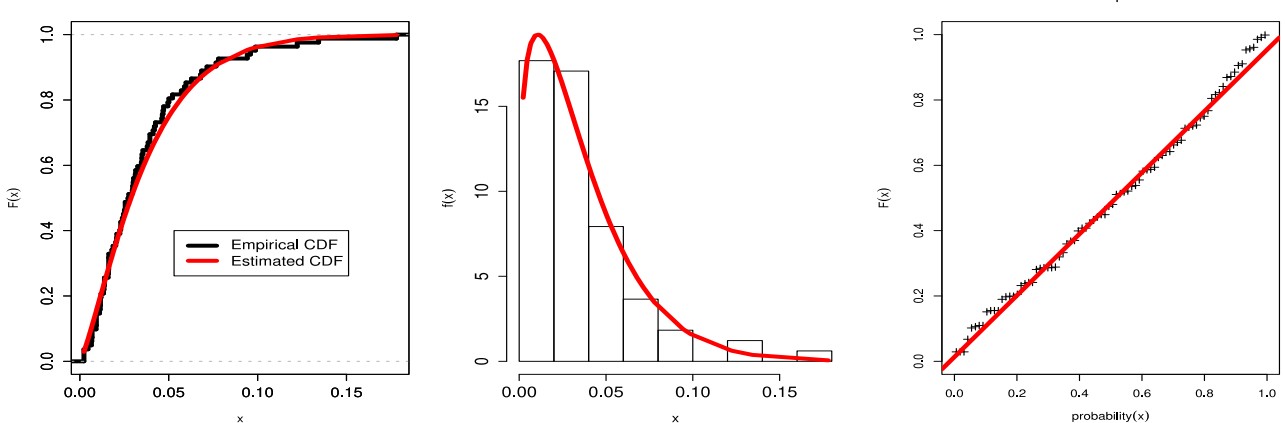

**Figure 13.** Plots of the estimated PDFs, CDFs, and P-P of the UEHLD of COVID-19 data.

Second, for the COVID-19 data, we determined the ML, MPS, and Bayesian estimates using the SELF of the UEHLD's parameters for COVID-19 data, which are listed in Table 11. We used the mentioned MCMC algorithm to generate 10,000 MCMC samples. The initial values of the unknown parameters were taken to be their MLEs in order to use the MCMC sampling procedure. For the COVID-19 data, Figure 14 draws the profile log-likelihood of the UEHLD for each parameter by fixing one parameter and changing the others. Figure 15 demonstrates the excellent behavior of the aforementioned data, since the two roots of the parameters are global maxima.

Figure 16 shows trace plots of the posterior distributions of the parameters to track the convergence of the MCMC outputs. Additionally, they display the histograms for the marginal posterior density estimates of the parameters based on 10,000 chain values and the Gaussian kernel, demonstrating how effectively the MCMC process converges. All

of the produced posteriors are symmetric with regard to the theoretical posterior density functions, as shown by the estimations.

**Table 11.** Estimates, SE, and CI of the UEHLD parameters for COVID-19 data.

| | ML | | | | Bayesian | | | |
|---|---|---|---|---|---|---|---|---|
| | Estimates | SE | Lower | Upper | Estimates | SE | Lower | Upper |
| $\varphi$ | 1.2515 | 0.1030 | 1.0496 | 1.4533 | 1.2511 | 0.0662 | 1.1236 | 1.3829 |
| $\delta$ | 29.3838 | 9.3197 | 11.1171 | 47.6505 | 29.4399 | 4.0952 | 21.2897 | 37.1251 |

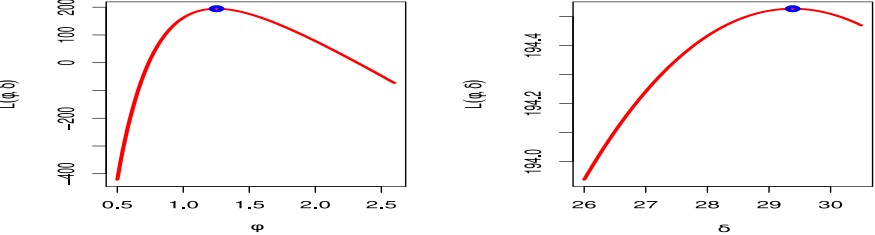

**Figure 14.** Profile likelihood for parameters of the UEHLD for COVID-19 data.

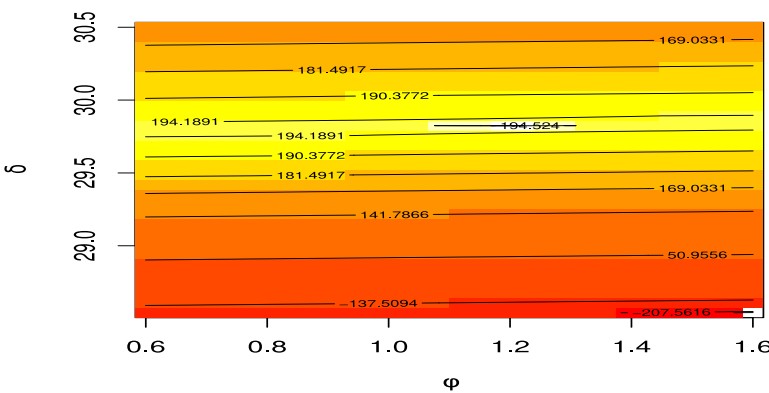

**Figure 15.** Contour plots of log functions with different UEHLD parameter values for COVID-19 data.

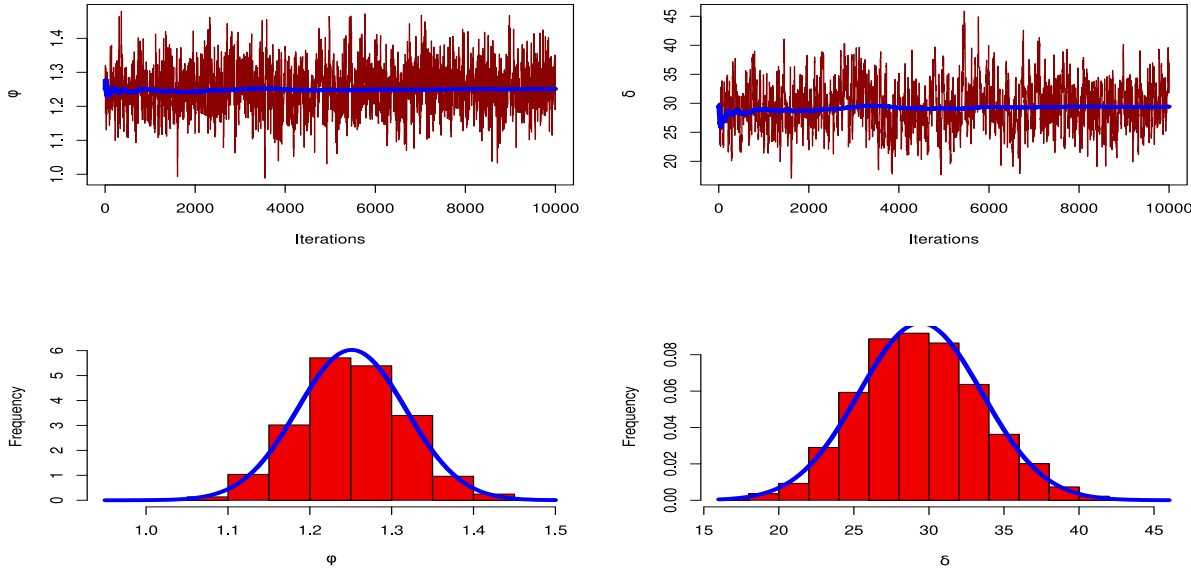

**Figure 16.** MCMC trace and posterior distribution for the UEHLD parameters using for COVID-19 data.

## 8. Summary and Conclusions

The unit-exponentiated half-logistic distribution, which is useful for modelling data on the unit interval, was proposed in this study as a result of an investigation into a suitable transformation. The mathematical properties of this distribution, such as moments, probability-weighted moments, incomplete moments, different entropy measures, and stress–strength reliability, were provided. The parameter estimators of the suggested distribution were established using the maximum likelihood, maximum product of spacing, and Bayesian methods. For the purpose of evaluating how well parameter estimates performed on finite samples, a thorough simulation study was included. The effectiveness of parameter estimation methodologies on finite samples was examined through a comprehensive simulation examination. We assessed the performance of point estimates in terms of their bias and MSE, while the interval estimates were investigated in terms of their length. We concluded that, in most cases, the smaller accuracy measures of the MPS estimates made them preferable to the ML estimates. The MSE reduced for each estimate as the sample size grew, demonstrating the consistency of the estimates. The length of the CI estimates based on the three techniques reduced as the sample size increased, indicating that the CI was the shortest. For clarification, the suggested distribution was practically applied to data on economic growth and tensile strength. Additionally, COVID-19 data analysis using British medical statistical data was supplied. In comparison to several new unit distributions and existing unbounded distributions, the experimental data showed that the proposed UEHLD distribution delivered a better outcome. It is important to note that, when analyzing the COVID-19 data, the result of our novel model produced superior measure values than the models shown by Abu El Azm et al. [37], Almetwally et al. [38], Hassan et al. [39], Liu et al. [40], Nagy et al. [41], Ahmadini et al. [42], and Mahmood et al. [43]. Additionally, estimates of the new model were taken into account for each dataset using various estimation techniques, with the exception of the first dataset, for which the MPS approach was not used due to its more equally distributed values. The UEHLD's log-likelihood was shown graphically together with the representations of the contour plots with varied parameter values. For the three datasets, histograms of the marginal posterior density estimates were provided along with plots of the posterior distributions of the parameters.

**Author Contributions:** Conceptualization, A.S.H., A.F. and A.A.; methodology, A.S.H., A.F. and E.M.A.; software, A.S.H. and E.M.A.; validation, A.S.H., A.F., A.A. and E.M.A.; formal analysis, A.S.H. and E.M.A.; investigation, A.F. and A.A.; resources, A.S.H. and A.A.; data curation, A.F. and E.M.A.; writing—original draft preparation, A.S.H., A.F. and E.M.A.; writing—review and editing, A.S.H., A.F., A.A. and E.M.A. All authors have read and agreed to the published version of the manuscript.

**Funding:** This research received no external funding.

**Data Availability Statement:** Data sets are available in this paper.

**Conflicts of Interest:** The authors declare no conflict of interest.

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
