# Peer review of "Bayesian and Non-Bayesian Inference for Unit-Exponentiated Half-Logistic Distribution with Data Analysis"

_applsci, doi:10.3390/app122111253_

Round 1

Reviewer 1 Report

This research paper is very interesting, but the author needs to be improved.

1. The Bayesian approach is based on subjective concepts, but the MLE approach is based on objective concepts. The author should not compare.

2. The author needs to write the research article in reference to standard journal format more than in Mathematical Statistics Exercises format.

3. The conclusion is not appropriate for this research article. Therefore, the author needs to revise and improve more. A good summary and conclusion needs to reflect the purpose of the research study.

Author Response

Response to Reviewer 1 Comments

First of all, we would like to express our sincere thanks and appreciation to the reviewers for their comments, which improved the paper. All the corrected parts are written with red color within the main text.

Comments

  1. The Bayesian approach is based on subjective concepts, but the MLE approach is based on objective concepts. The author should not compare.

Response 1:
Ok thanks, we delete these comparisons and add some new comments as given in subsection 6.2 and section 8

=======================

  1. The author needs to write the research article in reference to standard journal format more than in Mathematical Statistics Exercises format.

 Response 2:
Ok, thanks we write all references and their citations according to template journal.

=======================

  1. The conclusion is not appropriate for this research article. Therefore, the author needs to revise and improve more. A good summary and conclusion needs to reflect the purpose of the research study.

Response 2:
Ok, thanks we rewrite the conclusion and summary of the article to reflect the purpose of our study.

=======================

Best Regards

Reviewer 2 Report

In this paper, the authors devloped a unit-exponentiated half logistic distribution (UEHLD) model to model data which is on the unit interval. They examined the performance of the method to seveal data sets, including the tensile strength of polyester fibres and Covid 19 data in Britain. Here are my comments.

1. In page 2 line 54, the author may need to explain more about "lifetime" and show the detail reason of is importance to statistics.

2. The size and fonts of Figure 1 and 2 seem distorted. Please revise these parts.

3. The resolution of Figure 7 and 10 is low due to limited data. Can the authors increase the data points to obtain a more smooth color transition?

4. Table 3 and 4 can be placed in Supporting Information, since few conclusions can be summarized from the results.

5. The method description of "Real-Data Applications" can be simplified in the maintext, and the detail information can also be placed in Supporting Information.

6. Can the authors apply their methods into the data which may contain unit-interval and non-unit-interval? For example, in some countries the Covid-19 data are recorded on a weekday basis. Namely, the new cases found by hospitals on Saturday and Sunday will be reported on next Monday, which will cause pertubations to the data modelling. Thus it would be more important to develop a method that can handle both continuous data and discrete data.  

Author Response

Response to Reviewer 2 Comments

First of all, we would like to express our sincere thanks and appreciation to the reviewers for their comments, which improved the paper. All the corrected parts are written with red color within the main text.

Comments

  1. In page 2 line 54, the author may need to explain more about "lifetime" and show the detail reason of is importance to statistics.

Response 1:
Ok, we add more explaining about this issue in section 1 with red color. 

=======================

  1. The size and fonts of Figure 1 and 2 seem distorted. Please revise these parts.

Response 2:
Ok thanks, we re-plotted these figures

=======================

  1. The resolution of Figure 7 and 10 is low due to limited data. Can the authors increase the data points to obtain a more smooth color transition?

Response 3:
Done, thanks, we re-plotted these figures

=======================

  1. Table 3 and 4 can be placed in Supporting Information, since few conclusions can be summarized from the results.

Response 4:
Tables 3 and 4 have been organized and more comments are added in subsection 6.2

=======================

  1. The method description of "Real-Data Applications" can be simplified in the maintext, and the detail information can also be placed in Supporting Information.

Response 5:

Ok, thanks we add the following paragraph at the beginning of section 7:

“The effectiveness of the parameter estimator for the UEHLD for the three data sets under consideration is also assessed using the ML, MPS, and Bayesian techniques via a standard error and confidence interval length criteria measures. We get the estimators of the new model for three techniques for the data sets under consideration, with the exception of the first data set, for which the MPS approach is not employed because it has more equally values. For further clarification, the log-likelihood of the suggested model is supplied, along with examples of contour plots with various parameter values. We also provide plots of the posterior distributions of parameters as well as histograms for the marginal posterior density estimates for three data sets”

=======================

  1. Can the authors apply their methods into the data which may contain unit-interval and non-unit-interval? For example, in some countries the Covid-19 data are recorded on a weekday basis. Namely, the new cases found by hospitals on Saturday and Sunday will be reported on next Monday, which will cause pertubations to the data modelling. Thus it would be more important to develop a method that can handle both continuous data and discrete data.  

Response 6:

In this paper, unit continuous distribution has been introduced to illustrate useful quantities with values between 0 and 1. In this paper, we investigate an appropriate transformation to propose the unit-exponentiated half logistic distribution (UEHLD), which is also beneficial for modelling data on the unit interval. In the future, we will take into consideration your suggestion to develop a method that can handle both continuous data and discrete data.

 =======================

Best Regards

Round 2

Reviewer 2 Report

1. Figure 5 is still unclear and compressed at y axis.

2. Some texts should be formal or uniform. E.g. The x title of the third figure of Figure 3 should be Probability(x). The y title of the Figure 3 should all be F(x) or f(x) .

3. Thanks the author to improve the resolution of Figure 7. However the resolution of Figure 5, 8 is still low for publication.

4. The y axis of Figrue 14 and 15 seems compressed too.

Author Response

Answers to

Reviewers' Comments ( 2rd revised version )

 To the Editor-in-Chief

Applied Sciences

Manuscript Title:Bayesian and Non-Bayesian Inference for Unit Exponentiated Half Logistic Distribution with Data Analysis”

           Manuscript ID:  applsci-1942974

 Subject: Reply to the queries raised by the reviewers

Respected Editor

We considered the comments of the reviewers and revised the manuscript according to their suggestions. We have prepared the revision taking into account all these comments. We now answered all comments made by the referees. We thank the Editor and the referee for the constructive comments and hope that the revision is now appropriate for the Applied Sciences

Referee 2

First of all, we would like to thank you for thorough comments and remarks that significantly improved the former version. We hope that the revised version will match your expectation

Responses for Referee 2

  1. Figure 5 is still unclear and compressed at y axis.

Answer: Done

  1. Some texts should be formal or uniform. E.g. The x title of the third figure of Figure 3 should be Probability(x). The y title of the Figure 3 should all be F(x) or f(x) .

Answer: Done

Figure 3 discussed the 3D shapes of mean, ,   CV and ID for UEHLD

the third figure of Figure 3 discussed coefficient of variation (CV) with different values of parameters where x-lab is delta parameter  and y-lab is vatphi parameter  than the z-lab is cv value.

  1. Thanks the author to improve the resolution of Figure 7. However the resolution of Figure 5, 8 is still low for publication.

Answer: Thanks a lot Prof; In Figure 5, 8, we replotted these figures.

  1. The y axis of Figrue 14 and 15 seems compressed too.

Answer: All figures have been attached to journal of eps type with high quality